

# A Novel Method for Sea Surface Temperature Prediction using a Featural Granularity-Based ConvLSTM Model of Data-Knowledge-Driven

Mengmeng Cao[1,2], Kebiao Mao[2,3], Yibo Yan[4], Sayed M. Bateni[5], Zhonghua Guo[6]

[1]Key Laboratory of Geospatial Technology for the Middle and Lower Yellow River Regions (Henan University), Ministry of Education, Kaifeng 475004, China

[2]Institute of Agricultural Resources and Regional Planning, Chinese Academy of Agricultural Sciences, Beijing 100081, China.

[3]State Key Laboratory of Remote Sensing Science, Aerospace Information Research Institute, Chinese Academy of
Sciences. Beijing 100101, China

[4]College of Global change and Earth System Science, Beijing Normal University, Beijing 100875, China

[5]Department of Civil and Environmental Engineering and Water Resources Research Center, University of Hawaii at Manoa, Honolulu, HI 96822, USA

[6]School of Physics and Electronic-Engineering, Ningxia University, Yinchuan 750021, China

*Correspondence:* Kebiao Mao (maokebiao@caas.cn)

**Abstract:** Many data-driven methods predict sea surface temperature (SST) at specific locations using only the previous period SST values as predictors, which ignores the spatiotemporal dependencies of SST variability and the influence of multiple variables on SST patterns. Additionally, these methods have difficulty in capturing the feature dependencies involved in SST fluctuations, limiting the accuracy and horizon of SST predictions. This study proposes
a new medium- and long-term forecasting model to address these issues, which includes two sub-models: a featural granularity model and a data-knowledge-driven ConvLSTM prediction model. The former restacks the one-dimensional time-series of each variable into multidimensional feature variables using an adaptive granulation-based method. The latter integrates parameters that affect ocean dynamics and thermodynamics, along with pixel-to-pixel similarity, to achieve partition predictions. The multidimensional feature variables were fed into the ConvLSTM
model to exploit the feature- spatiotemporal patterns for predictions. Experiments conducted in three sea areas of the western Pacific and Indian Oceans indicate that the use of featural granularity can enhance the ability of the prediction model to capture dynamic characteristics in the time domain and internal dependencies of the features and extend prediction horizons. The combination of knowledge-driven and study area segmentation concepts can help the prediction model better capture the unique features and dynamics of the local area, further improving prediction
accuracy. Validation against observations and cross-comparisons with baseline models in three different sea areas for the prediction of monthly SST with lead times ranging from 1 to 120 months demonstrate that the proposed model can generate consistent and more accurate regional SST predictions. The differences between predicted and observed values range from -0.7 to 0.7K, with an RMSE of approximately 0.3 to 0.57K for SST predictions. This developed model provides a promising approach for medium- and long-term sea surface temperature forecasting, which can be
easily adapted to other ocean parameter prediction tasks.



## 1 Introduction

Sea Surface Temperature (SST) refers to the temperature of the upper layer of the ocean's surface, which represents the temperature of the water in direct contact with the atmosphere. SST is a crucial variable within the global ocean-atmospheric system, playing a vital role in the exchange of heat, moisture, momentum, and gases between the ocean and the atmosphere (Cao et al., 2021; Xue et al., 2022). Changes in SST can have a significant impact on the global climate and ecosystems, potentially leading to extreme events (e.g., typhoons, droughts, and floods) (Bentamy et al., 2017; Minnett et al., 2019; Xiao et al., 2019a). Therefore, future projections of SST are of significance in the early warning of extreme events and understanding climate dynamics.

Currently, methods for predicting SST can be divided into two main categories. One is physics-based numerical models. It uses a series of complex mathematical equations coupled with the physical laws of the ocean to simulate ocean dynamics and predict SST (Aparna et al., 2018; Liu and Fu, 2018; Stockdale et al., 2006). For example, Noh et al. (2002) improved the prediction accuracy of SST by embedding a new ocean mixed layer model into an ocean general circulation model. Krishnamurti et al. (2006) predicted global SST seasonal anomalies by constructing a super-ensemble based on 13 state-of-the-art coupled atmosphere-ocean models. Some agencies including the European Centre for Medium-Range Weather Forecasts (ECMWF) and the National Centers for Environmental Prediction Command (NCEP) also provide coupled model forecasts based on ensemble techniques, where ECMWF and NCEP can forecast various parameters 10-15 days and 380 hours in advance, respectively. Although SST forecasts based on numerical models can achieve good accuracy over large spatial regions, they usually involve complex external data and multiple assumptions for model initiation and time integration (Patil and Deo, 2018; Xu et al., 2020). Their performance is determined by the coupling and data assimilation mechanisms, and furthermore, since various parameters are simultaneously predicted, it is challenging to precisely tune a single parameter.

The other category is data-driven models, which can learn the latent characteristics from historical data and further predict SSTs using the learned patterns (Su et al., 2018; Su et al., 2021; Xiao et al., 2022; Xin et al., 2020). Some data-driven models have been used, such as Markov models (Xue and Leetmaa, 2000), support vector regression (Imani et al., 2017), empirical canonical correlation analysis (Collins et al., 2004; Tang et al., 2000), linear regression (Kug et al., 2004), empirical orthogonal functions (Neetu et al., 2011), and artificial neural networks (ANNs). These approaches have excellent tractability and are especially advantageous when information regarding the physical mechanisms of real-world processes is inadequate (Xiao et al., 2019b; Xu et al., 2020). Among these data-driven approaches, ANNs tend to be more popular because they can fully explore the complex patterns hidden in the data and model them (Aguilar-Martinez and Hsieh, 2009; Wu et al., 2006). For example, Xiao et al. (2019a) developed a hybrid prediction model based on LSTM and AdaBoost methods for accurately predicting location-specific SSTs. Patil and Deo (2017) used wavelet neural networks to predict the SST at six selected sites in the Indian Ocean and obtained good prediction results. However, they are limited to predicting SST in a few locations and cannot simultaneously provide spatial information similar to the physics-based numerical models. Patil and Deo (2018) proposed a data-driven model with a very large number of individual ANNs (20.5902 million networks) running simultaneously for basin-scale SST predictions. However, the large number of networks may affect the running speed of the computer



and the promotion of the model. Furthermore, changes in SST occur through advection and diffusion, indicating an interaction between the SST of a specific location and that of its neighboring points (De Bézenac et al., 2019). Therefore, the independent prediction of SSTs at each location disregards the interaction between SSTs at different locations, potentially impacting the accuracy and validity of the prediction.

Convolutional long short-term memory (ConvLSTM) is capable of learning dependent information over time while incorporating spatial features, making it effective for processing time series data with both temporal autocorrelation and spatial characteristics. Xiao et al. (2019b) successfully achieved precise SST prediction in the East China Sea utilizing the ConvLSTM model, which effectively captures the spatial and temporal correlations inherent in SST data. Hao et al. (2023) studied the impact of model structure (different parameter settings) on the performance of the ConvLSTM and ST-ConvLSTM (an improved version of ConvLSTM) models in predicting SST and found ConvLSTM has good performance in predicting SST in the South China Sea. In addition, some researchers have also considered the influence of historical SST information at locations near the prediction point by adding convolutional layers with regional information extraction capabilities to the neural network. However, these models, which utilize SST observations as inputs, do not capture the internal dependencies of specific periodic features involved in SST fluctuations (Shao et al., 2022; Yang et al., 2018; Yu et al., 2020). Furthermore, the actual SST variation is influenced by the interaction of multiple parameters. Many data-driven SST forecasting methods tend to treat SST as a single parameter and solely analyze its time series, ignoring the interplay among different parameters in terms of dynamics and thermal effects.

To bridge these gaps, this paper proposes a novel method for SST prediction based on granular computing and the ConvLSTM model of data-knowledge-driven. This method fully considers the influence of the driving factors of SST changes, greatly exploits the spatiotemporal information of SST sequences, and extends the prediction horizons. Validation against observations and cross-comparisons with baseline models in three different sea areas were performed to demonstrate the reliability and suitability of the proposed prediction method for medium- and long-term SST forecasting.

## 2 Methodology

Fig. 1 shows the working mechanism of the proposed granulation-based ConvLSTM Model of Data-Knowledge-Driven, which involves three stages: data pre-processing based on featural granularity model and expert knowledge, ConvLSTM training, and accuracy assessment. In the first stage, we utilize the relevant theories and research results of previous experts on the dynamic changes of SST as an expert knowledge base as a knowledge-driven, thereby incorporating human practical experience and wisdom. Specifically, 13 meteorological and oceanic parameters that reflect ocean thermal and dynamical processes, as well as sea-air interactions, were selected as input predictors for SST prediction by integrating the physical mechanisms of SST variability. By quantifying the similarity between the pixels within the study area, the study area was divided into different sub-regions, and different parameters were selected for each sub-region as predictors for SST prediction. Finally, the one-dimensional time series of each predictor is extracted pixel by pixel. Each one-dimensional time series was adaptively segmented according to its trend characteristics to generate unequal-length temporal granules and was restacked into multidimensional feature variables.

In the second stage, the feature variables were fed into the constructed ConvLSTM model to achieve the prediction of the feature variables associated with the SST. In the third stage, the SSTs were obtained by applying the degranulation

process to these predictions. The accuracy of the proposed prediction model was evaluated through a comparison with other models, namely, Convolutional Gated Recurrent Unit (ConvGRU), Convolutional Neural Network (CNN), Deep Learning Neural Network (DLNN), Bi-Directional Long Short-Term Memory (Bi-LSTM), and Fully Connected Long Short-Term Memory network (FC-LSTM). Each step is described in the following subsections. In Section 3, we apply the method to three different sea areas and made detailed analysis.

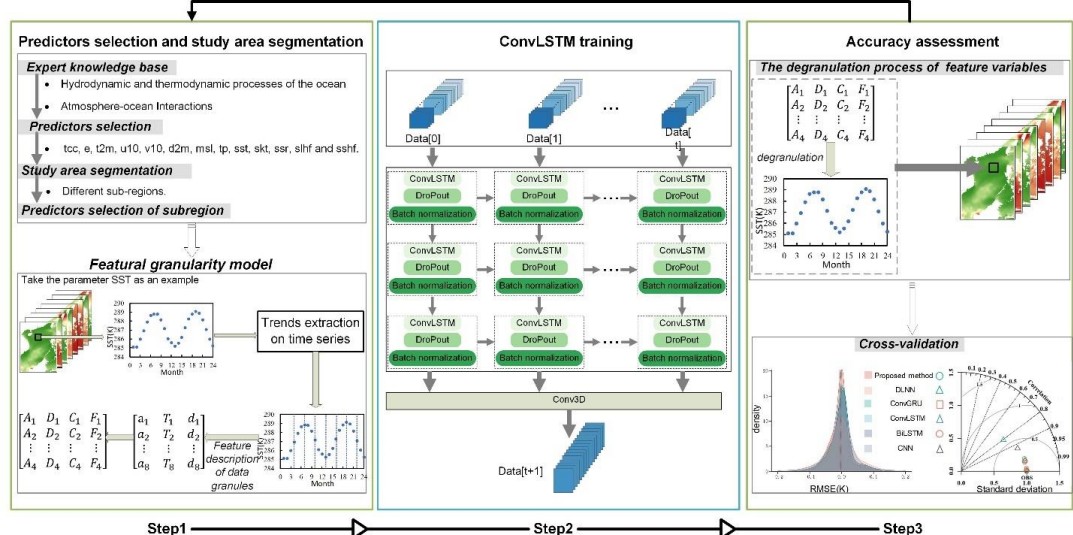


**Fig. 1** Visual representation of the proposed model. The cross-validation in Step 3 is based on the RMSE values of predictions obtained using the proposed model and baseline models.

## 2.1 Predictors selection and study area segmentation

Changes in SST result from the interaction of various meteorological and oceanic variables, with dynamic and
thermal interactions between different variables (Shao et al., 2022). Therefore, incorporating surface parameters that reflect these processes may be more effective for predicting SST. Variables such as total cloud cover (tcc), evaporation (e), 2m temperature (t2m), 10m u-component of wind (u10), 10m v-component of wind (v10), 2m dewpoint temperature (d2m), mean sea level pressure (msl), total precipitation (tp), sea surface temperature (sst), sea skin temperature (skt), surface net solar radiation (ssr), surface latent heat flux (slhf) and surface sensible heat flux (sshf)
can influence SST variability patterns by affecting ocean thermal and dynamical processes as well as sea-air interactions. Clouds modulate SST by reflecting incoming sunlight, while evaporation can affect SST by influencing sea-air interactions. Sea-air temperature difference, relative humidity and wind speed can modulate changes in latent heat fluxes by affecting ocean evaporation, which in turn affects changes in SST. Additionally, evaporation and precipitation cause changes in the salinity and temperature of the ocean mixed layer, affecting the buoyancy and
density at the ocean surface. This leads to mixing and convection between surface and subsurface oceans, which in





turn affects SST. Wind stress forcing can further intensify vertical processes exchanging surface and subsurface water (e.g., mixing, vertical advection, and turbulence). Hence, the 13 aforementioned meteorological, seawater states, and dynamics variables were utilized as an expert knowledge base. The input predictors for predicting SST will be derived from this knowledge base.

Although SST in different regions of the ocean is influenced by these factors, their relative importance can vary significantly from one region to another due to unique regional characteristics and interactions. Therefore, by quantifying the similarity between the pixels within the study area, the study area was divided into different sub-regions, and different parameters were selected for each sub-region as predictors for SST prediction. The process involves grouping pixels with similar meteorological and oceanic conditions, which can be achieved by using

correlation-based methods to create co-occurrence networks. Co-occurrence networks can be constructed by calculating the correlation coefficient matrix for every two pixels in the study area and selected statistically significant and strongly correlated pixels (i.e., correlation coefficient > 0.8 and p-value < 0.01) (Jiao et al., 2020). Fig. 2(a) shows co-occurrence networks generated from the matrix of predictors for all pixels in study area I (The South China Sea). The nodes in the network represent pixels, and the edges connecting two nodes indicate the correlation between them.

Each color corresponds to a distinct module, with pixels within each module sharing similar meteorological and oceanic conditions. Through the identification of pixel positions and associated variables within each module, we can divide each study area into distinct subregions and identify the key variables influencing SST changes within these subregions. For example, the South China Sea is divided into two sub-regions, i.e., region 1 and region 2 (Fig. 2(b)). The key variables that affect the SST change in region 1 are sst, d2m, skt, tcc, t2m, sshf, msl, u10, tp and ssr, and the

key variables that affect the SST change in region 2 are sst, ssr, msl, skt, t2m, slhf, e, u10, v10, sshf and d2m.

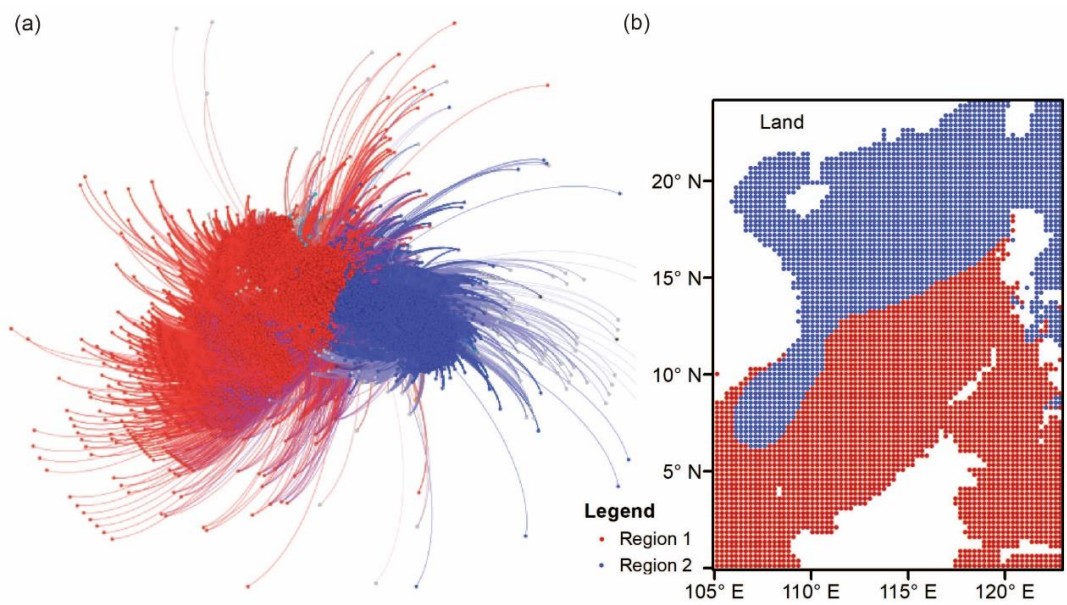

**Fig.2** (a) Co-occurrence networks generated from the matrix of predictors for all pixels in study area I. (b) The study



area I is divided into two sub-regions.

The drivers of SST variability in different regions are not identical, and too many predictors can increase the complexity of the subsequent prediction model, thereby reducing the prediction accuracy. While the Co-occurrence network diagram can visually identify factors that affect SST dynamics in a region, it does not inherently determine the importance or relative influence of each variable. When many variables are identified, it becomes challenging to prioritize or select the most critical ones. Therefore, we further used the random forest algorithm, which has the ability to measure the importance of variables, to select the important predictors in different regions. By integrating the

insights from the Co-occurrence network with the variable importance scores derived from the random forest analysis, we were able to identify and prioritize the most significant factors influencing SST, thereby improving the efficiency and accuracy of our prediction model. Taking study area I as an example, Fig. 3 shows the importance ranking of the 13 predictors in regions 1 and 2 based on the random forest algorithm and the prediction errors using different numbers of predictor variables after ranking by importance. The prediction accuracy of the model increases and then decreases

as the number of input predictors increases, both for region 1 and region 2. For region 1, the model has a high accuracy of prediction when eight variables are selected: sst, skt, t2m, sshf, msl, u10, tp and ssr. For region 2, the model has a high accuracy of prediction when nine variables are selected: sst, skt, t2m, slhf, e, u10, v10, sshf and d2m. Following this, a comparison will be made between the outcomes derived from the co-occurrence network and the random forest analysis. The common variables identified will then be utilized as predictor variables in regional models aimed at

predicting SST.

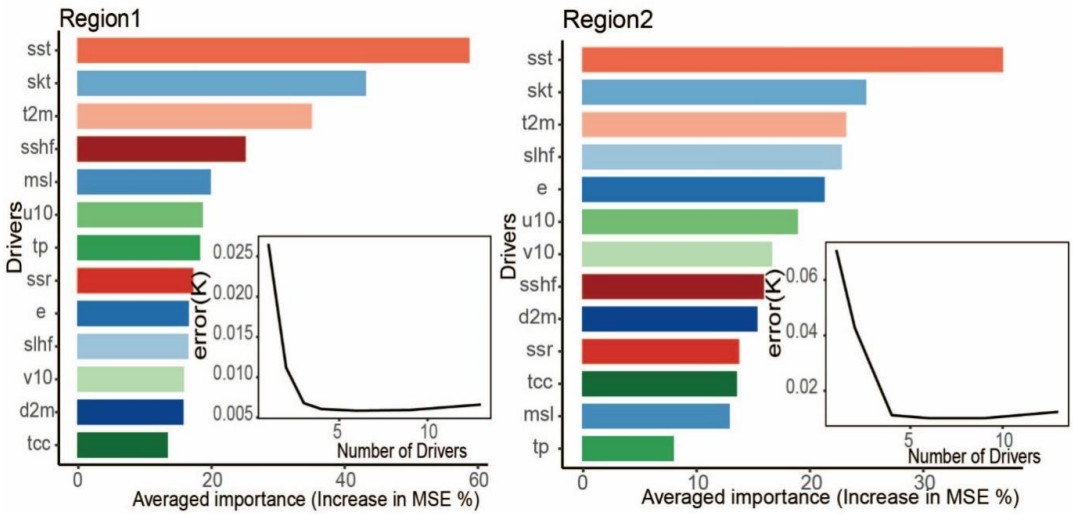

**Fig. 3** Results of the importance of drivers based on random forest and the prediction errors using different numbers of predictor variables after ranking by importance (study area I)

**2.2 Formation of multi-series features based on featural granularity model**

Information granules refer to sets of related data or information that are considered as a unit. In general, periodic or partially periodic time series can be divided into a series of information granules (i.e., temporally continuous trend





segments) according to their tendency features. These information granules can be approximated by stretching a series of templates in the horizontal and vertical directions (Cao et al., 2023). Therefore, we first granulated the one-dimensional time series for each variable at each pixel into data granules within various tendency feature. Specifically,

we partition information granules based on the monotonicity and concavity-convexity of the target variable series $X$ ($X = \{x_1, x_2, ..., x_n\}$). The monotonicity and concavity-convexity of $X$ are represented by its first-order dynamic $X'$ ($X' = \{x'_1, x'_2 ..., x'_{n-1}\}$) and second-order dynamic $X''$ ($X'' = \{x''_1, x''_2 ..., x''_{n-2}\}$), respectively.

$$x'_t = x_{t+1} - x_t \quad (t \in [1,2,...,n-1] ) \quad\quad\quad (1)$$

$$x''_t = x'_{t+1} - x'_t \quad (t \in [1,2,...,n-2] ) \quad\quad\quad (2)$$

If the monotonicity and concavity-convexity of $X$ change at the data point $x_t$, then $x_t$ is considered as the splitting point of $X$. That is, if $x'_t * x'_{t+1} < 0 \cup x''_t * x''_{t+1} < 0$, the monotonicity and/or concavity-convexity of the data point $x_t$ changes, and thus, the target variable series $X$ can be divided into $\{x_1, x_2, ..., x_t\}$ and $\{x_{t+1}, ..., x_n\}$. In addition, to address the noisy characteristic of the target variable series or small variations in concavity-convexity that may lead to dense partitioning, two constraints are added to each variable series through data analysis and multiple

experiments: $|t - 1| > \theta_j \cap |x_t - x_1| > \emptyset_j$, where $\theta_j$ and $\emptyset_j$ are the thresholds for the $j$th variable series. In this manner, each variable series is granulated into information granules with various trend characteristics.

Subsequently, we used quarter-circle sinusoids as templates and applied appropriate horizontal and vertical scale stretches to approximate the information granules. We then introduced a 3-D feature space, represented by amplitude (a), template type (T1-T4), and duration (d), to characterize the original granules, as illustrated in Fig. 4(a-b).

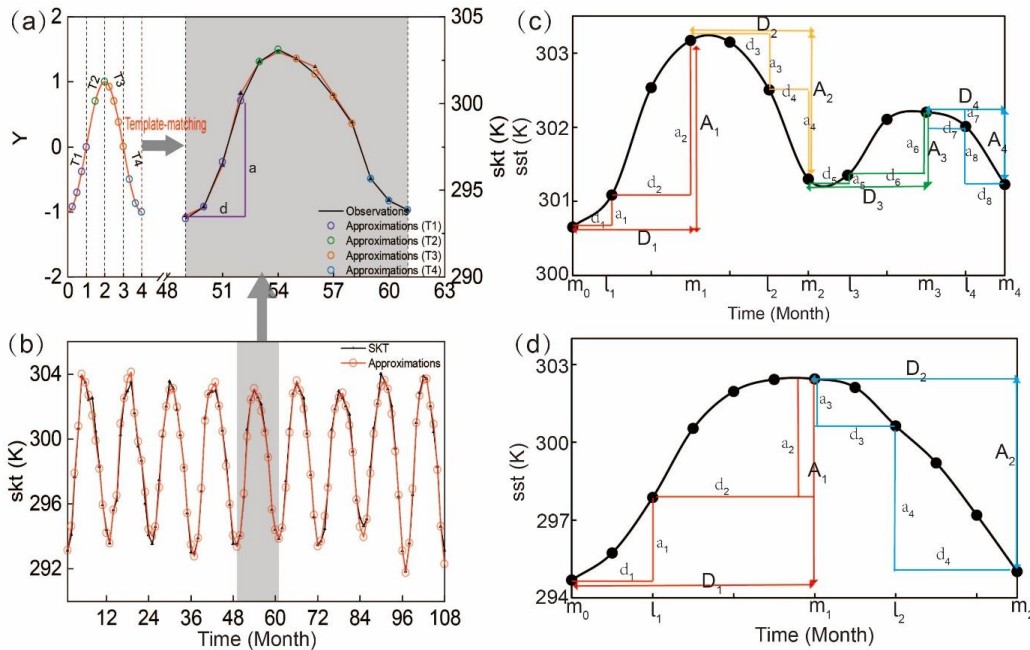


**Fig. 4** (a) Four quarter-period sinusoids used as templates and approximate SKT values obtained through template





matching. (b) Original variable values of pixels at 106.11°E, 18.84°N and their approximation through the combination of stretched templates. (c) Formation of multi-series features for SST variables at 107.10°E, 0.37°N. (d) Formation of multi-series features for SST variables at 107.10°E, 17.45°N.

Using the 3-D feature space directly to describe the original granules formed by 1-dimensional time series of each variable at each pixel leads to information redundancy, thereby diminishing the accuracy of subsequent predictive modeling and escalating computational complexity. Moreover, the examination of unequal-length information granules generated by various variables at each pixel reveals that the information granules of an uptrend are usually concave followed by convex or first convex and then concave, whereas the downward trend is first convex and then 205 concave or concave followed by convex. Considering these two factors, we combined adjacent unequal-length information granules with different concavities and convexities but the same monotonicity to form new information granules. Throughout the duration of the study, it was observed that nearly all variables exhibited a notable periodicity at each pixel, with a recurring cycle of 12 months. We identified the number of new information granules ($h$) during each cycle to construct $h$ sets of new durations ($D$) and amplitudes (A), as shown in Fig. 4(c-d), where the sign of A 210 denotes the monotonicity of the granules. Subsequently, the 3-D feature space consisting of amplitude, template type, and time domain for each variable at each pixel is transformed into a 4*h-D feature space ($[A_1, D_1, F_1, C_1, A_2, D_2, F_2, C_2, \cdots, A_h, D_h, F_h, C_h]$) consisting of the newly constructed amplitude (A), duration (D), curvature ($C$), and fluctuation (F) features. C and $F$ can be calculated using the amplitude and duration of the granules. An example of an SST variable for this calculation is shown in Fig. 4(c-d). Suppose $m_0, m_1, m_2, m_3$ and $m_4$ are 215 the times of monotonicity division, i.e., $m \in \{i | x_t' * x_{t+1}' < 0\}$. $l_1, l_2, l_3$ and $l_4$ are the times of concavity-convexity division, that is, $l \in \{i | x_t'' * x_{t+1}'' < 0\}$. The 16-D feature space of the granules between $m_0$ and $m_4$ can be represented by $[A_1, D_1, F_1, C_1, A_2, D_2, F_2, C_2, A_3, D_3, F_3, C_3, A_4, D_4, F_4, C_4]$, where $A_t = x_{m_t} - x_{m_{t-1}}$, $D_t = m_t - m_{t-1} + 1$. $F_t$ and $C_t$ can be calculated using Equations 3 and 4, respectively. Since the new feature space is going to be used as input for machine learning, we need to ensure that the feature space of these variables at each pixel has the 220 same number of elements. After analyzing each pixel in three distinct study areas, it was determined that approximately 9.47% exhibited inconsistencies in the feature space elements of the predictors. Subsequently, through a series of experiments, the inconsistent pixels were addressed by filling the corresponding positions in the feature space of each variable with a value of 0 to maintain consistency in the number of elements.

$$C_t = \frac{l_t - m_{t-1}}{m_t - m_{t-1}} \tag{3}$$

$$F_t = \frac{|x_{l_t} - x_{m_{t-1}}|}{|x_{m_t} - x_{t-1}|} \tag{4}$$

### 2.3 Construction of ConvLSTM model

ConvLSTM is a neural network model that combines CNNs and LSTM networks (Shi et al., 2015). Its basic idea is to combine the recurrent neural network structure of LSTM with the convolutional operation of CNN to simultaneously capture spatial and temporal features of a sequence. In ConvLSTM, each LSTM unit is modified to a 230 convolutional LSTM unit, which combines convolutional operation with the state update and gating operations of the



LSTM unit to model and predict spatiotemporal sequence data. The structure of ConvLSTM units is shown in Fig. 5.

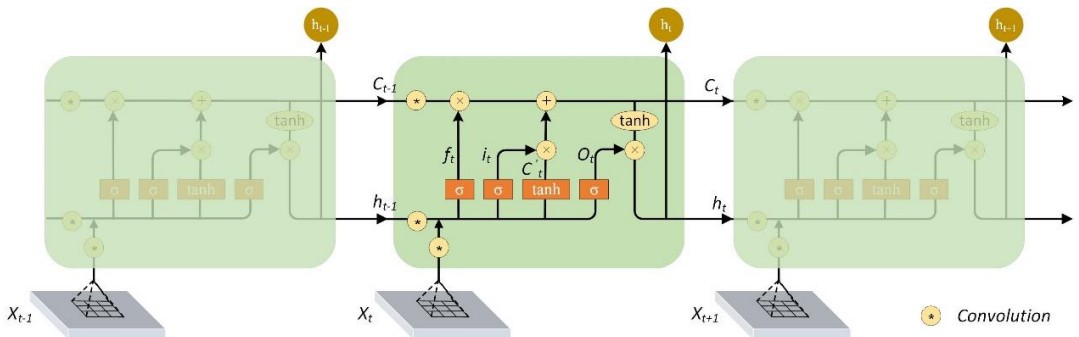

**Fig. 5** Structure of ConvLSTM units

$X_t$ is the input of the ConvLSTM in which the gates $f_t$, $i_t$, and $O_t$ and the candidate cell state $C'_t$ are controlled

by ($X_t$, $h_{t-1}$). The cell state $C_t$ is updated using $f_t$ and $i_t$. $O_t$ determines the amount of information propagated to the time step $t$+1. These gates consist of a sigmoid fully connected neural network layer and a point-wise multiplication operation. The working mechanism of these gates and the information flow can be represented as follows:

$$f_t = \sigma(W_{xf} * X_t + W_{hf} * h_{t-1} + W_{cf}°C_{t-1} + b_f) \tag{5}$$


$$i_t = \sigma(W_{xi} * X_t + W_{hi} * h_{t-1} + W_{ci}°C_{t-1} + b_i) \tag{6}$$

$$C'_t = \tanh(W_{xc} * X_t + W_{hc} * h_{t-1} + b_c) \tag{7}$$

$$C_t = f_t°C_{t-1} + i_t°C'_t \tag{8}$$

$$O_t = \sigma(W_{xo} * X_t + W_{ho} * h_{t-1} + W_{co}°C_{t-1} + b_o) \tag{9}$$

$$h_t = O_t°\tanh(C_t) \tag{10}$$

where $X_t$ is the input at time t, $h_{t-1}$ and $C_{t-1}$ are the hidden state and cell state at the previous time step, and W and b are the weight and bias parameters of the corresponding gates. $*$ denotes the convolution operator. $°$ denotes element-wise multiplication, and σ and tanh are the sigmoid and hyperbolic tangent activation functions, respectively.

Due to interactions within the global climate system, it is important to model not only the spatial relationships among neighboring regions but also to capture the spatial dependencies of any given region (Mu et al., 2019). However,

a single convolutional layer can only account for spatial dependencies in close proximity, as it is limited by the size of its kernels. Furthermore, the variation of SST is influenced not only by the most recent month but also by interannual climate variations. Thus, it is necessary to consider both short-term and long-term temporal information in the entire prediction model. Therefore, to achieve high-precision prediction results, we need to stack multiple ConvLSTM layers with different numbers of kernels and kernel sizes to derive a prediction model.

The hyperparameters of the prediction model (e.g., the number of ConvLSTM layers, number of kernels and kernel size, and dropout rate) must be tuned to improve its performance. In this study, 10% of the data from the training dataset was randomly selected to tune the hyperparameters of the prediction model. We first tested a relatively simple model with only one ConvLSTM layer and then generated new models by conducting several operations, including





changing one or two hyperparameter settings, setting more ConvLSTM and Convolutional layers, adjusting the order

of layers, or replacing part of the network with more complex components. Three indexes including the mean absolute error (MAE), the determination coefficient ($R^2$), and the root mean square error (RMSE) were used to evaluate the prediction performance of these models with different parameter settings, and models with good prediction performance were used as the seeds to start a new round of the best model searching. Finally, the optimal parameters of the ConvLSTM network architecture can be determined until the prediction performance is no longer improved.

After testing the performance of different optimizers, we used the Adam optimizer for training. Through this process, we constructed a 4-layer deep neural network model for predicting SST feature variables, as shown in Fig. 6. The feature variables constructed in section 2.2 of t continuous time steps are taken as the input data. These input data are then passed through three ConvLSTM layers and one Conv3D layer, respectively, to output $i$ 16-feature variable images corresponding to the SST variable at time steps t+1 to t+i. The four convolutional layers have filter sizes of

7x7, 3x3, 5x5, and 7x7x7, with 50, 50, 50, and 16 filters, respectively. A dropout mechanism and a regularization layer are added after each ConvLSTM layer, with dropout values of 0.5, 0.5, and 0.2, respectively.

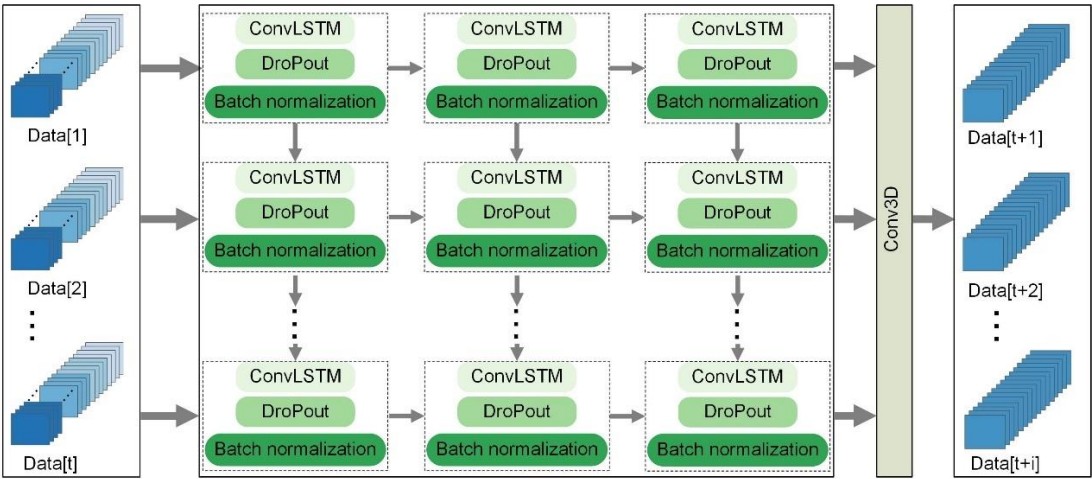

Fig. 6 The architecture of the proposed ConvLSTM network.

### 2.4 Transform the obtained feature variables to SST

The final predictions of the ConvLSTM model are feature variables. The SSTs can be obtained by transforming the predictions, which is referred to as the degranulation of the obtained feature variables. The transformation formula is contingent upon the templates utilized to transform SSTs into multi-series features. In this study, quarter-cycle sinusoids were used as templates, and therefore, the SSTs were determined using the sinusoidal formula:

$$x_i = a\sin\left(\frac{\pi i}{2d} + \emptyset\right) + b \tag{11}$$

where $x$ is the $i$th SST value of the template, and $b$ is the value of its starting SST point. The parameter $\emptyset$ is determined by the type of matching template, for example, concave increasing and decreasing templates are used when $\emptyset$ is 0 and $\pi/2$, respectively. $a$ and $d$ denote the amplitude and duration of the matching templates, respectively,



which can be calculated using the final prediction of the feature variable.

## 3 Experimental results and discussion

### 3.1 Study area and data

The spatial extent of study area I is 0-25°N and 105-125°E, as shown in Fig. 7. It is bordered by two broad continental shelves to the north and south, with a maximum depth of about 5,000 m in the east-central part of the sea (Li et al., 2007; Pan et al., 2013). This study area is located in the South China Sea, one of the largest tropical marginal seas in the western Pacific Ocean, and is surrounded by China, Vietnam, Malaysia, Indonesia, and the Philippines

(Kuo et al., 2000; Wang et al., 2021). The South China Sea is of critical ecological importance, with significant petroleum reserves, important international trade routes, and rich fishery resources (Li et al., 2021).

To enhance the assessment of the proposed SST prediction model, Study areas II and III were also included for SST prediction. The spatial extents of these two study areas are 20 ~ 35°S and 75-98°E, and 45 ~ 60°S and 60-90°E, respectively, located in the southern Indian Ocean, far from land. These regions were chosen due to their geographical

overlap with high latitudes (60-90°S), mid-latitudes (30-60°S), and low latitudes (0-30°N). Additionally, they encompass the tropical zone (23.5°N-23.5°S), the southern temperate zone (23.5-66.5°S), and the southern cold zone (66.5-90°S) in terms of temperature zones. Furthermore, the three study areas vary in their distances from the shoreline. These factors contribute to a diverse range of spatial changes in SST within the study areas, enhancing the evaluation of the SST prediction model's effectiveness.

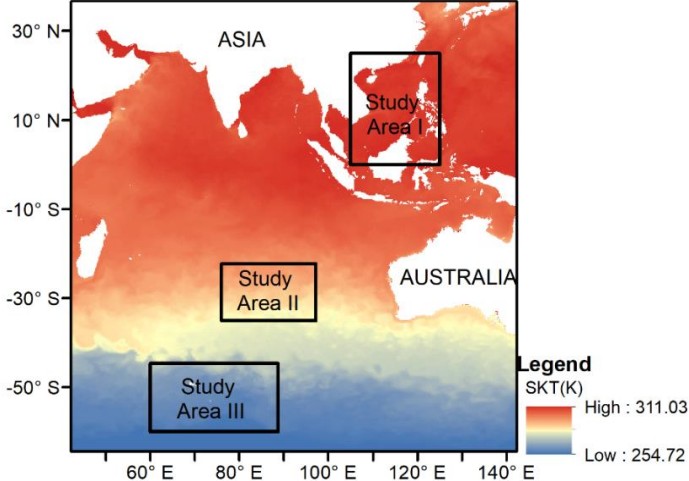


**Fig. 7** Location map of the study area

In this study, we used monthly meteorological parameters with a spatial resolution of 0.25° from 1959 to 2021 to predict SST. These data can be obtained from website (https://cds.climate.copernicus.eu/cdsapp#!/dataset/reanalysis-era5-single-levels-monthly-means, last access: January 10, 2022). These monthly meteorological parameters were

obtained from the ERA5 dataset generated by ECMWF, which are generated by combining simulated and observed



data all over the world (Hersbach et al., 2020). The ERA5 dataset is available from 1959, but it is not sufficient to support high-precision SST predictions using only these 756 months of data. To expand the temporal coverage of these data, we used the CMIP5 monthly data on single levels from ECMWF for 1850–2005, which can be obtained from website (https://cds.climate.copernicus.eu/cdsapp#!/dataset/projections-cmip5-monthly-single-levels, last access: January 10, 2022). Unlike the ERA5 dataset, which assimilates upper-air and satellite data, the CMIP5 monthly dataset is generated by coupled Global Circulation Model (GCM) simulations that inform the Intergovernmental Panel on Climate Change (IPCC) reports. In order to maintain consistency between the two datasets, a regression model was developed to establish a relationship between ERA5 and CMIP data, with the CMIP data subsequently calibrated through regression matching (Meng et al., 2021). The data from 1850 to 2011 were used as the training and validation sets for the prediction model, and the data from 2012 to 2021 were used for prediction to further validate the accuracy of the prediction model.

## 3.2 Experiment setup

In this study, we used the feature variables constructed in Section 2.2 of t continuous time steps as input data to predict the feature variable images constructed based on SST for the next 10 consecutive steps. To ensure that the ConvLSTM model we constructed achieved the best predictions for different study areas, we analyzed the variation in the magnitude of the loss function with the number of epochs in the ConvLSTM model for each study area's predictions and selected the number of epochs in which the loss functions remained nearly constant. Study areas I, II, and III were partitioned into 2, 2, and 3 sub-areas, respectively, using the pixel similarity evaluation method outlined in Section 2.1. The number of epochs predicted for the two sub-areas of study area I, the two sub-areas of study area II, and the three sub-areas of study area III were set to 264, 192, 208, 222, 215, 198, and 246, respectively. We also examined the variations in $R^2$, MAE, and RMSE of the ConvLSTM model for each sub-region prediction with the timestep, and the optimal timestep for prediction of the two sub-regions in study area I, the two sub-regions in study area II, and the three sub-regions in study area III were set to 18, 21, 16, 22, 15, 23, and 26, respectively.

## 3.3 Results and discussion

### 3.3.1 Effect of study area segmentation on prediction performance of feature variables

To evaluate the impact of regional partitioning on the final result, we used feature variables derived from 13 predictive indicators as inputs for the constructed ConvLSTM model to predict the target variables (based on SST-constructed feature variables) for three study areas and their seven subdivided areas. It can be found that compartmentalized prediction performs much better than the overall prediction for all prediction horizons. Fig. 8 shows the predictive performance of the constructed model for 1 to 10 steps ahead of predictions for the entire study area I and its sub-regions. This is due to the fact that different regions within the study area exhibit different spatial patterns and temporal dynamics, which can be better captured by dividing the area into smaller sub-regions based on the temporal and spatial characteristics of the predictors within the study area.



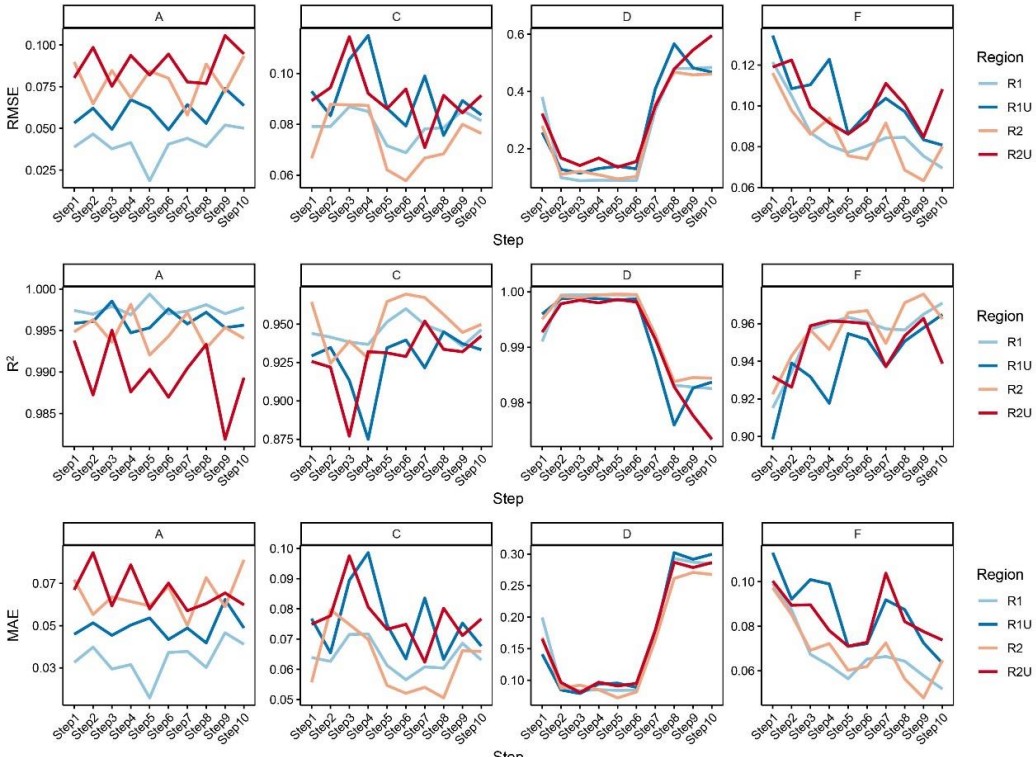

**Fig. 8** the proposed model's predictive performance for 1 to 10 steps ahead predictions for the study area I and its sub-
regions. R1 and R2 denote the results of individual predictions for regions 1 and 2 of study I, respectively. R1U and
R2U represent the results of both regions when the overall prediction is made.

### 3.3.2 Effect of knowledge-driven on prediction performance of feature variables

To evaluate the impact of meteorological and oceanic parameters that affect ocean dynamics and thermodynamics
on the predictive performance of the model, we conducted a comparative analysis of three distinct inputs to evaluate
their predictive efficacy on target variables within seven sub-regions. The three inputs were as follows: (1) predicting
the target variables for seven sub-regions using solely the SST-based constructed feature variables, (2) predicting the
target variables for seven sub-regions using feature variables constructed based on 13 indicators that affect ocean
dynamics, and (3) predicting the target variables for seven sub-regions using feature variables constructed from various
indicators selected separately for each sub-region, based on the contribution analysis conducted in subsection 2.1.
Figs.9 and 10 show the comparison of the prediction results for the different inputs (Due to space limitations, it is
difficult to present all the results for all seven sub-regions. Therefore, we only show the results for the two sub-regions
divided by study area I). The inclusion of multivariate factors improves the predictive performance of the model,
enabling it to consider the dynamic balance between various variables and providing a better fit with the real marine
environment. Therefore, for each sub-regions, using feature variables constructed from 13 indicators affecting the




ocean dynamics pattern as inputs results in higher prediction accuracy for each horizon compared to using only SST-based feature variables. Moreover, this finding suggests that data-driven forecasting methods should consider the joint effect of multiple variables rather than treating each variable individually. Additionally, we observe that upon scrutinizing the indicator contributions of each subregion and selecting inputs based on their significant importance

contributions, the prediction accuracy of the subregions shows no notable diminution across all prediction horizons. Notably, at several prediction horizons, the accuracy surpasses that of the model employing all feature variables as inputs. This could be because not all of the 13 indicators are highly correlated with the SST changes in these sub-regions, while some variables have weak explanatory power for the SST changes or even have a negative impact on the prediction results. In this case, using more indicators does not necessarily improve the prediction accuracy, as they

do not provide additional information and may even introduce noise or interference.

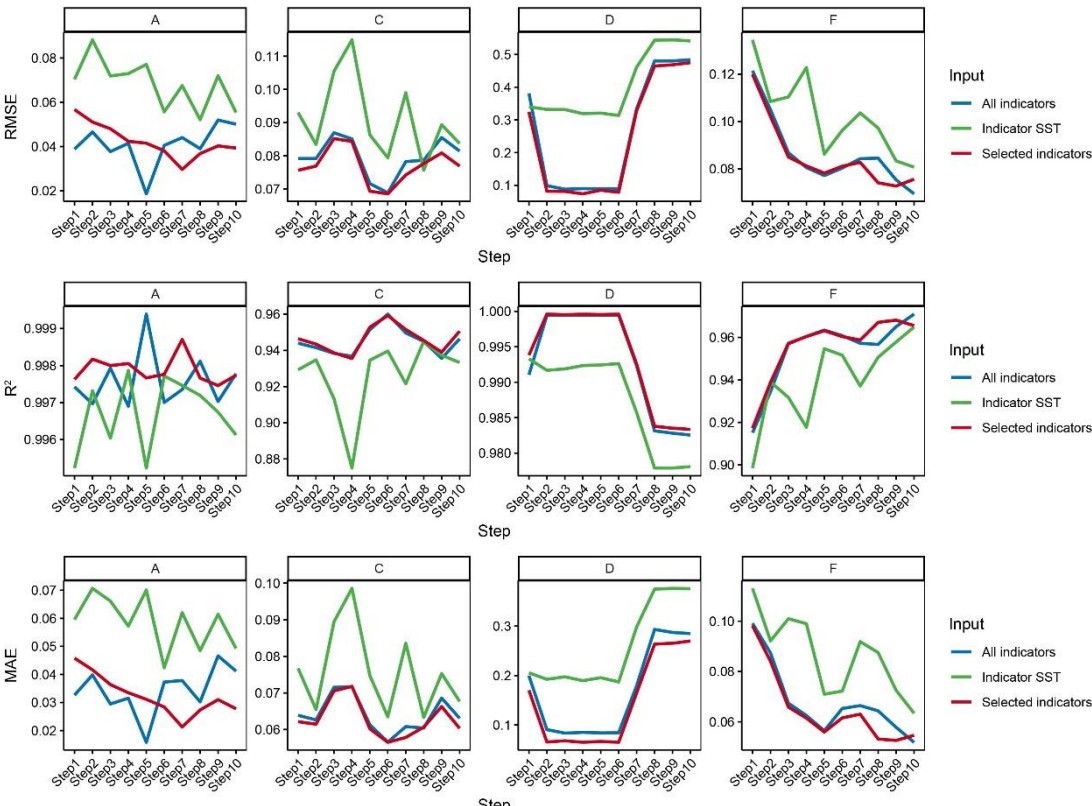

**Fig.9** Prediction results of feature variables under three different input conditions in sub-region 1 of study area I



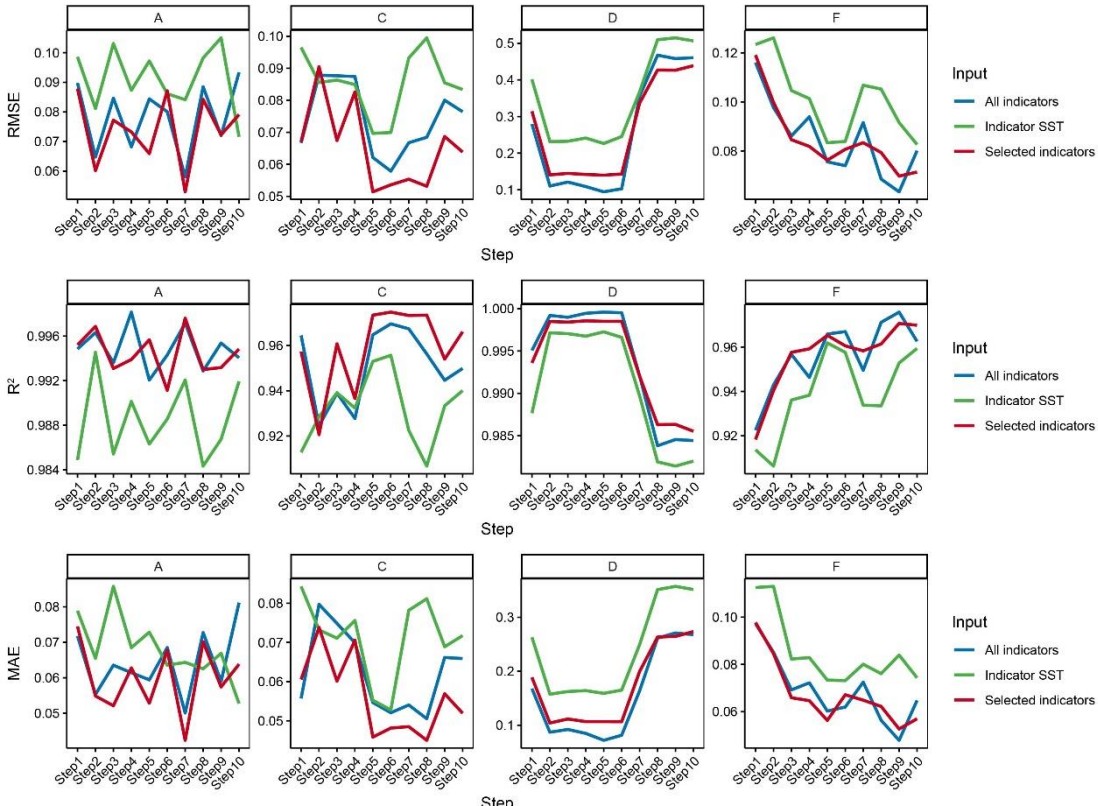

**Fig.10** Prediction results of feature variables under three different input conditions in sub-region 2 of study area I

### 3.3.3 Effectiveness of Model on SST prediction


The SSTs were obtained by applying the degranulation process to the predicted feature variables. We analyzed the prediction results of the test samples from various perspectives and found that the predicted SSTs and their spatial distribution are in good agreement with the observed SST images. The discrepancies between the predicted and observed SSTs are primarily within the temperature range of -0.7 to 0.7 K. Fig.11 illustrates the predicted SSTs for

the 12 months of 2009 in study area I. The first column represents the observed SST images, the second column is the predicted SST images, the third column is the differences between the observed and predicted SST images, and the fourth column is a histogram of these differences. Due to space limitations, we randomly selected the predicted results for 2009 for display. It is clear that the predicted SSTs and their spatial distribution for all 12 months of 2009 are consistent with the observed SST images. The differences between the predicted and observed SSTs fluctuate within

the range of -0.7 to 0.7 K for over 94.3% of the pixels throughout the January-December period. Although Fig. 11 indicates that some images contain values with significant errors, the histogram of the differences illustrates that these pixels with large errors are typically very few, averaging no more than 30 pixels per image. These pixels are often located at the land-sea interface, which may be due to the large spatial resolution of the data used in this study and the





fact that areas close to the shore often consist of mixed pixels. The small number of mixed pixels in the images may
lead to insufficient feature learning of the proposed model for these pixels. In addition, the distribution of errors in
each image does not exhibit a specific pattern. However, comparing the error distributions with the distributions of
each indicator selected for the corresponding month reveals that regions with relatively large errors often coincide
with higher wind speeds. Due to the lack of a clear regularity in wind speed variation, constructing an accurate
multiseries characterization of wind speed for this study is challenging. Consequently, the prediction accuracy of SST
in regions with large wind speed variations is inevitably affected.

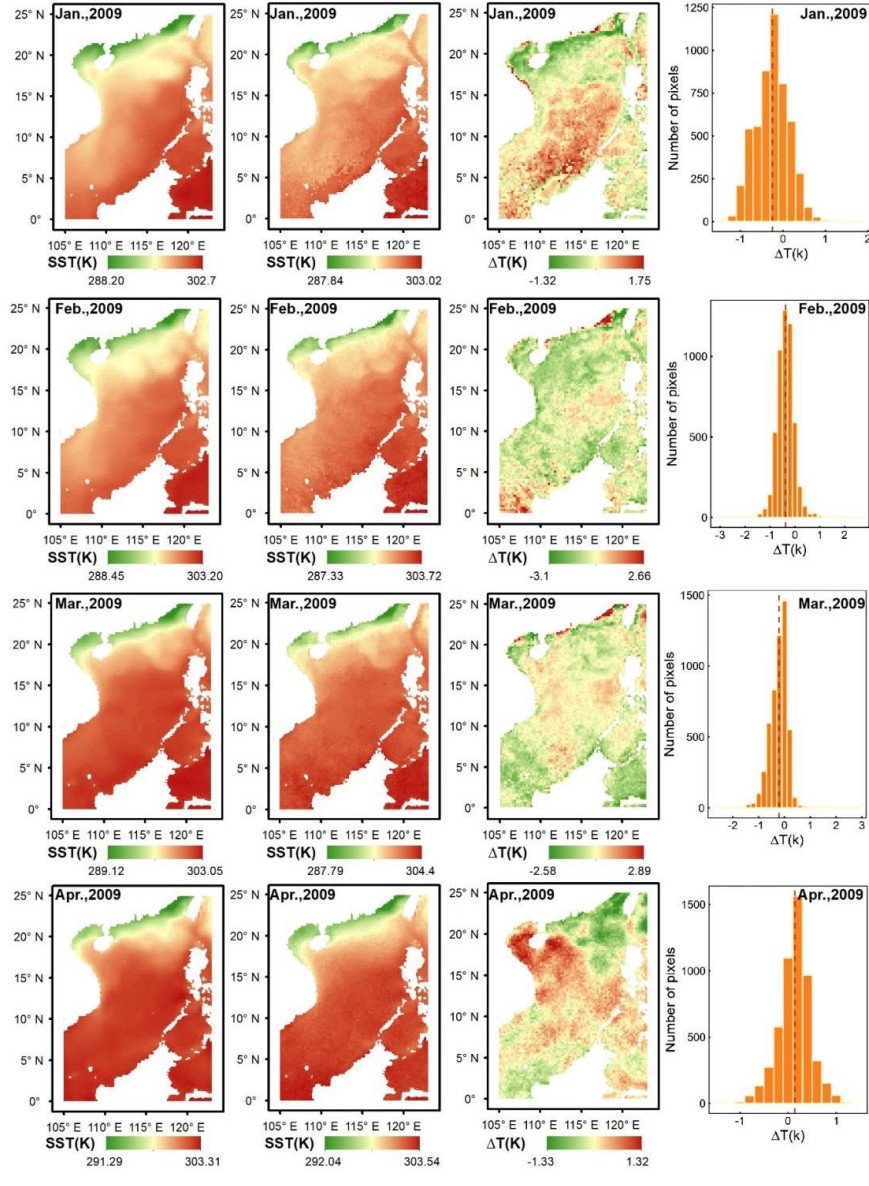





**Fig.11** The predicted SSTs (first column), observed SSTs (second column), and spatial distribution (third column) and



statistics (fourth column) of prediction errors for study area I in 2009

### 3.3.4 Accuracy comparison between the proposed model and the baseline models

To further evaluate the prediction performance of proposed method for SST, we separately predicted the SSTs for the years 2012-2021 using our method and five widely baseline models for SST prediction. The optimal hyperparameters for the baseline models were determined using a grid search method. The proposed prediction model was established considering units of feature segments rather than individual SST values. Thus, the proposed prediction method could obtain 120 months ahead predictions of SST in the study area after performing one prediction and degranulation. To fairly compare the SSTs predicted by the baseline models and our proposed method, we set the prediction length of one run of the baseline models to 120. However, we found that the prediction accuracy of the five baseline models was very low when their prediction length was set to 120. In order to make a comparison between our predicted models and the baseline models, we experimented with the prediction length of the baseline model and found that these baseline models had the highest accuracy when it was set to 48. This also implies that these baseline models are difficult to use for longer predictions. Therefore, we first evaluated the prediction performance of our proposed method for 120 months of SST from 2012-2021, and then compared our first 48 predictions with the 48 predictions of each baseline model to judge the performance of our proposed method.

Due to limited space, it is difficult to show the forecast results for all of these 120 months. Therefore, we present the forecast results for the year with the relatively lowest total forecast accuracy (2021) in this 10-year period. If there is high prediction accuracy in 2021, it proves that the predictions of other years are also reliable. Fig. 12 shows the predicted SSTs for each month of the year 2021 within study area I. The first column represents the observed SST images, the second column is the predicted SST images, the third column is the difference between the observed and predicted SST images, and the fourth column exhibits the density plot depicting the disparities between the predicted and observed values for the respective months in 2021 and 2020. Our proposed methodology demonstrates the ability to forecast SST distribution for the year 2021, with discrepancies between predicted and observed values typically falling within the range of -1 to 1 K. The forecast accuracy for 2021 is slightly lower than that for 2020, with February, March, May, June, October, and December having lower forecast accuracy for the SST in 2021 than that for 2020. In contrast, the forecast accuracy for the other months is similar to that of 2020, indicating that the forecasts for other years are reliable.













**Fig.12** The predicted SSTs (first column), observed SSTs (second column), and spatial distribution (third column) and statistics (fourth column) of prediction errors for the study area I in 2021

We conducted a comparative analysis between our initial 48 predictions for the three study areas and the 48



predictions generated by each baseline model, as depicted in Figs. 13-15. The modeling process in this study focused on key periodic characteristics, such as fluctuation amplitude and duration. As a result, the forecasting results closely

matched the fluctuation trends and amplitudes of real data across different time scales. Additionally, the proposed model integrates the influence of multiple variables and sub-regions during the modeling process, leading to the lowest RMSE and superior performance. In the monthly SST predictions over up to 48 months for study areas I, II, and III, the discrepancies between the predicted and observed SSTs fluctuated within the ranges of -0.6 to 0.54 K, -0.3 to 0.4 K, and -0.4 to 0.35 K, respectively, with RMSEs of approximately 0.49 K, 0.2 K, and 0.21 K, respectively.

Although both the Bi-LSTM model and FC-LSTM model do not consider the spatial dependence of SST, their prediction accuracy still outperforms that of the DLNN and CNN models. In the monthly SST predictions for months 1-48 in study area I, the RMSEs between the predicted and observed values for BILSTM, FCLSTM, CNN, and DNN are approximately 0.68 K, 0.75 K, 0.87 K, and 0.93 K, respectively. For study area II, the RMSEs for BILSTM, FCLSTM, CNN, and DNN are approximately 0.25 K, 0.28 K, 0.33 K, and 0.37 K, respectively. In study area III, the

RMSEs for BILSTM, FCLSTM, CNN, and DNN are approximately 0.32 K, 0.40 K, 0.51 K, and 0.55 K, respectively. This is because the Bi-LSTM and LSTM models are good at long-term dependence modeling. Furthermore, in this study, we modeled each pixel when making predictions for these study areas, so these two models have relatively high accuracy. The experimental results also indicate that the prediction performance of ConvGRU for SST lies between that of BiLSTM and LSTM, with BiLSTM outperforming LSTM. In the monthly SST predictions for months 1-48 in

study areas I, II, and III, the RMSEs between ConvGRU's predicted and observed values are approximately 0.73 K, 0.26 K, and 0.4 K, respectively. BiLSTM extends LSTM by processing the sequence in both forward and backward directions. This bidirectional approach allows BiLSTM to leverage information from both past and future states, providing a more comprehensive understanding of the temporal context. This often leads to improved prediction accuracy because BiLSTM can capture dependencies that a unidirectional LSTM might miss. ConvGRU's simpler

GRU units have fewer parameters and gates compared to LSTM and BiLSTM, making it computationally more efficient but potentially less powerful in capturing long-term dependencies. The DLNN model exhibits the worst performance among all the models. This may be due to the fact that CNN can learn hierarchical representations, which can capture the spatial dependency of SST. A typical CNN architecture consists of layers with progressively larger receptive fields, enabling the network to learn increasingly complex features from low-level to high-level features.

DLNNs typically use a shallower architecture and are less capable of building hierarchical representations.





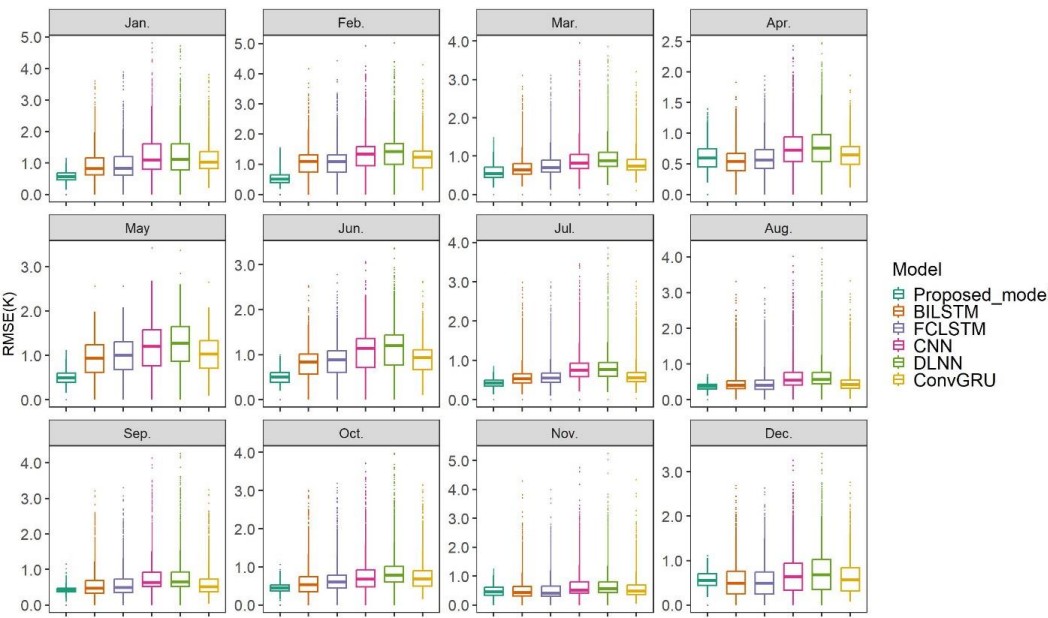

**Fig.13** RMSE of SST predictions for the years 2012-2021 within study area I, obtained using the baseline models and the proposed model.

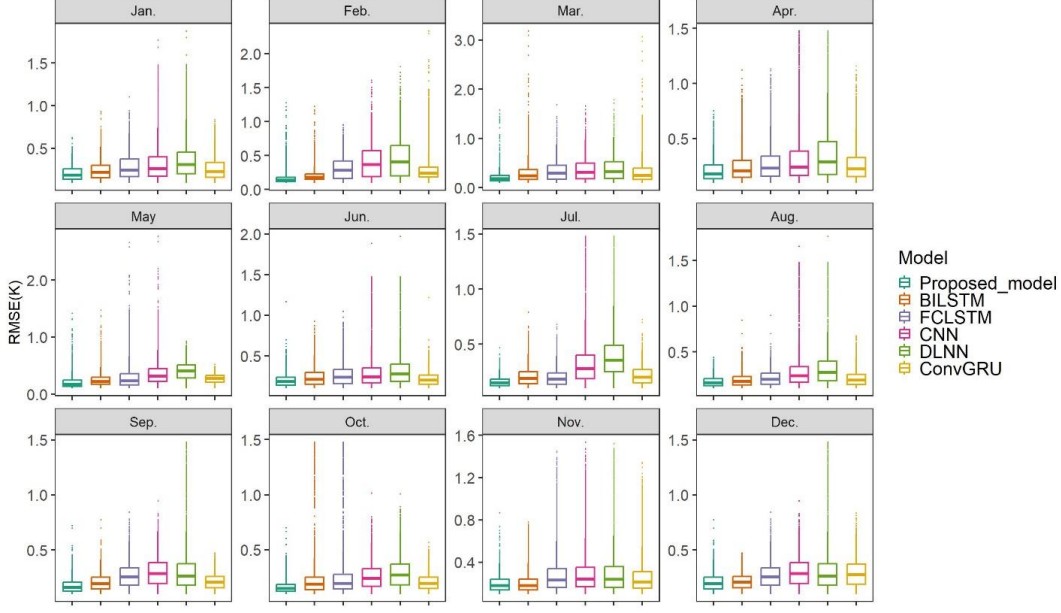

**Fig.14** RMSE of SST predictions for the years 2012-2021 within study area II, obtained using the baseline models and the proposed model.



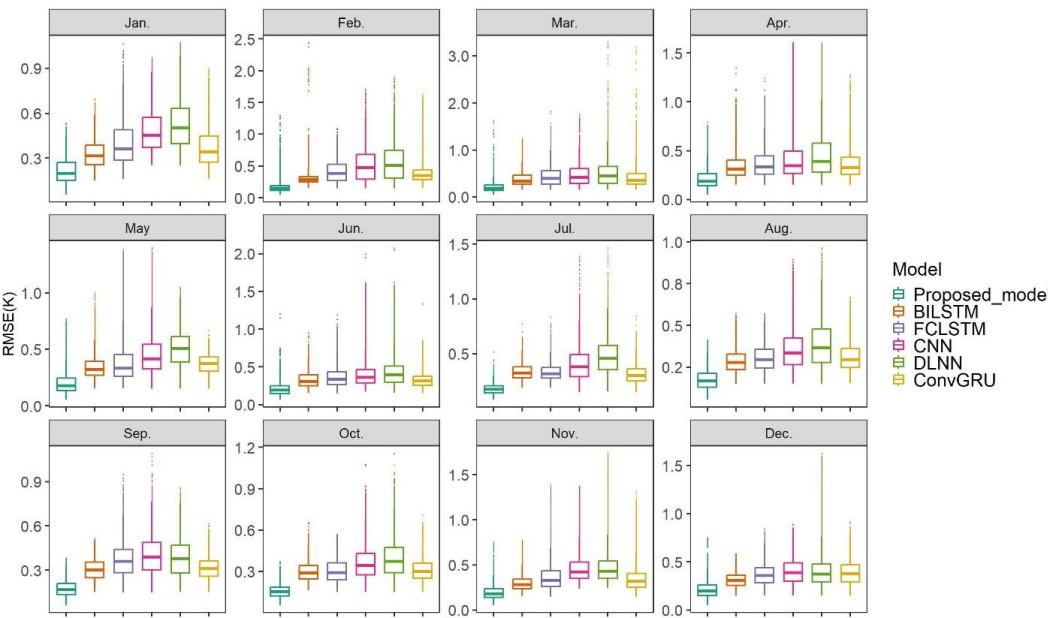

**Fig.15** RMSE of SST predictions for the years 2012-2021 within study area III, obtained using the baseline models and the proposed model.

## 4 Conclusions

In this study, we proposed a novel method for SST prediction based on granular computing and the ConvLSTM model of data-knowledge-driven. The model considers multiple influencing factors in ocean dynamics and thermodynamics processes and can fully utilize spatiotemporal information of image sequences to improve the accuracy of SST prediction. Our experiments showed that the combination of knowledge-driven and study area segmentation concepts can enable the development of customized prediction models that are tailored to the specific characteristics of the study area and the available knowledge. This can improve the interpretability and understanding of the prediction model and further improve the accuracy of the prediction. In addition, the introduction of the idea of feature granularity allows machine learning models to fully capture the dynamic characteristics in the time domain and internal dependencies of the features and extend the prediction horizons. The comparison of different statistical indicators and different perspectives also shows that the proposed model has strong robustness and generalization ability and has great potential in the time series prediction of SST. In the monthly SST prediction of the South China Sea region (study area I) for up to 120 months, the difference between the predicted SSTs and the observed SSTs fluctuates within -0.7-0.7 K, with an RMSE of about 0.57K. Similarly, in study areas II and III, the monthly SST predictions over the same 120-month period exhibited a variance between predicted and observed temperatures ranging from -0.5 to 0.5 K, with an RMSE ranging from 0.3 to 0.5 K.



Our research provides a new idea and method for SST prediction. The extended prediction horizons provided by the proposed method are also a significant improvement over existing model. This is important for decision-makers who require accurate and reliable forecasts for planning and management purposes, especially in areas such as fisheries, marine transportation, and climate change mitigation. It is worth noting that the proposed method is not without
limitations. The use of the granular computing model we constructed requires that the sequence of variables be periodic or aperiodic, which may limit the applicability of the proposed method in some cases. In the future, we can further improve the predictive performance of SST by coupling numerical and machine learning models to enable the models to better account for ocean dynamics and thermodynamic processes.

**Code and Data availability:** The codes for conducting the analyses can be downloaded from https://doi.org/
10.1080/17538947.2023.2260779 (Cao et al., 2023) and https://doi.org/10.5281/zenodo.14759549 (Cao et al., 2025). The data on which this article is based are available in https://doi.org/10.1002/qj.4174 (Bell et al., 2021) and https://doi.org/10.1175/BAMS-D-11-00094.1 (Taylor et al., 2012).

**Author contribution:** MC and KM: Data curation, Investigation, Methodology, Writing. YY, SB and ZG: Conceptualization, Methodology, Formal analysis, Investigation.

**Declaration of competing interest:** The authors declare no competing financial interests.

**Acknowledgments:** The authors extend their heartfelt appreciation to ECMWF for providing the climate reanalysis product, Google for their invaluable contribution of the open-source machine learning framework, and TensorFlow for offering the sophisticated machine learning library that served as the cornerstone for conducting these experiments. This research is funded by the Second Tibetan Plateau Scientific Expedition and Research Program (STEP) project
"Dynamic monitoring and simulation of water cycle in Asian water tower area" (no. 2019QZKK0206), the National Key R&D Program of China (no. 2021YFD1500101), the Open Fund of the State Key Laboratory of Remote Sensing Science (no. OFSLRSS202201), Ningxia Science and Technology Department Flexible Introduction talent project (no. 2021RXTDLX14), and Fengyun Ap-plication Pioneering Project (no. FY-APP-2022.0205).

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
