# Peer review of "A Novel Method for Sea Surface Temperature Prediction using a Featural Granularity-Based ConvLSTM Model of Data-Knowledge-Driven"

_EGUsphere, 2025_

## Referee Comment (RC3)

I challenge the value of proposed SST forecast.

In the ocean, the monthly SST, as the predicted variable of this study, has strong seasonality. If we use the SST of, for example, Jan of 2008, as the 'forecast' of Jan of 2009, we will find unexpectedly small differences between them, generally <2 degree. This is called 'persistence' forecast, but using the seasonality information. We may call it 'climatological forecast'. Fig. 1 shows an example of such a climatology forecast.

My point is the proposed method does not exceed such a simple 'forecast', with higher forecast errors, making the forecast operationally useless. When given SST for a deep learning to learn, it only learns the seasonal pattern, not subtle year-to-year SST changes. That is the reason why many studies predict SST anomaly, not SST itself.

[Figure]

Fig. 1 Example of SST climatological forecast

Besides, the manuscript is poorly written.
(1) Although in the abstract, the authors claim that 120-month forecast is obtained, but no results were shown in the manuscript.
(2) Section 3.3.4 is written in a non-informative and verbose way.
(3) https://doi.org/10.1080/17538947.2023.2260779 this DOI in the Code and Data Availability is in-accessible.
(4) The analysis in Section 3 is superficial and pointless.

---

## Community Comment (CC3)

Dear Shouyi Zhong,

Thank you very much for your careful review of our manuscript and for your insightful comments. We appreciate the time and effort you have dedicated to providing valuable feedback. The replies are presented below.

1. Comparison with Random Forest Model

In our earlier work, we combined LSTM and RF models and observed that the standalone RF model yielded lower prediction accuracy compared to LSTM and the LSTM-RF hybrid model (as shown in Figure 1). Based on these findings, we selected LSTM as a baseline model for accuracy validation in our current study. During the precision validation phase, we did not specifically compare the results of RF against proposed model in this study, as LSTM was already designated as a benchmark model within our study.

Given the relevant literature proposing the use of ConvGRU for SST prediction, ConvGRU has been added as a baseline model in our comparative analysis to provide a more comprehensive evaluation of predictive performance. This allows for a direct comparison between ConvGRU and our proposed ConvLSTM model, enriching the assessment of our methodology.

2. Determination of Step Size (Timestep)

To illustrate our optimization process, we use the two South China Sea sub-regions (Region 1 & Region 2) as representative examples:

We used the feature variables constructed in paper of t continuous time steps as input data to predict the feature variable images constructed based on SST for the next 10 consecutive steps. To ensure that the ConvLSTM model we constructed achieved the best predictions for regions 1 and 2, we analyzed the variation in the magnitude of the loss function with the number of epochs in the ConvLSTM model for region 1 and region 2 predictions and selected the number of epochs in which the loss functions remained nearly constant (as shown in Fig. 2). The number of epochs predicted for regions 1 and 2 was set to 264 and 192, respectively. We also examined the variations in R2, MAE, and RMSE of the ConvLSTM model for region 1 and region 2 predictions with the timestep (as shown in Fig. 3), and the optimal timestep for region 1 and region 2 predictions was set to 18 and 21, respectively. This approach accounts for localized ocean dynamics, ensuring our ConvLSTM captures

spatially heterogeneous temporal dependencies.

Thank you again for your constructive feedback and welcome any further suggestions you may have.

[Figure]

Fig. 1. Prediction errors for different prediction horizons (1-12 months) using the LSTM-RF, LSTM and RF methods at the five locations.

[Figure]

Fig.2 Variation of model loss with the number of epochs for different region prediction.

[Figure]

Fig.3 The relationship between prediction accuracy and timestep of feature variables in regions 1 and 2 based on the ConvLSTM model.

---

## Community Comment (CC4)

Dear Dr. Añel,

Thank you profoundly for your rigorous review and for guiding us toward full compliance with the journal's policies; we deeply apologize for any concerns caused and are truly grateful for your dedication to transparent scholarship.

We immediately downloaded the original Zenodo attachment (https://doi.org/10.5281/zenodo.14759549) and verified that it contains code files when accessed via our database interface, as shown in the screenshot fig. 1. To ensure universal accessibility, we proactively uploaded a new ZIP-format code package (https://doi.org/10.5281/zenodo.15714288) with identical content and will updated the manuscript to prioritize this direct-access option.

[Figure]

Fig.1 Original Zenodo file opened via database interface, showing active code files

Regarding data availability, we have transferred all public dataset links (originally in Section 3.1) into the "Code and Data Availability" section, adding explicit URLs for all datasets used. The revised section now states:

**Code and Data availability:** The codes for conducting the analyses can be downloaded from https://zenodo.org/records/15714288. The data on which this article is based are available in https://cds.climate.copernicus.eu/datasets/reanalysis-era5-single-levels-monthly-means?tab=overview (previous interface: https://cds.climate.copernicus.eu/cdsapp#!/dataset/reanalysis-era5-single-levelsmonthly-means) and https://cds.climate.copernicus.eu/datasets/projections-cmip5-monthly-single-levels?tab=overview (previous interface: https://cds.climate.copernicus.eu/cdsapp#!/dataset/projections-cmip5-monthly-single-levels).

Your expertise has been invaluable in improving this work. We are committed to upholding GMD's policies rigorously and appreciate your guidance. Should further refinements be needed, we will address them immediately.

With utmost respect and appreciation,

Mengmeng Cao

---

## Author Comment (AC1)

Dear Wenbin Tang,

Thank you for your insightful question regarding the impact of the information granule merging operation. This step is a crucial design choice in our feature engineering pipeline, fundamentally aimed at transforming fragmented, noisy trend segments into meaningful representations of complete trend cycles while significantly enhancing computational efficiency and model interpretability. Below, we elaborate on its specific impacts on the construction of the 4×h-D feature space and its role in capturing key dynamic characteristics of the time series.

The initial granulation based on monotonicity and concavity-convexity often results in a large number of fine-grained information granules due to noise or minor fluctuations (e.g., small convex/concave variations within an overall rising/falling trend). While these granules locally approximate the series, they introduce significant redundancy and fail to represent holistic trend behavior. Critically, we observed that natural periodic patterns (e.g., annual SST variations) consistently manifest as sequences of concave-convex or convex-concave sub-cycles within a sustained monotonic phase. For instance, a complete warming trend typically comprises an initial concave segment (accelerating increase) followed by a convex segment (decelerating increase). Merging adjacent granules with identical monotonicity but opposing concavity-convexity consolidates these physically related sub-cycles into unified "macrogranules." Each macrogranule now represents a complete acceleration-deceleration phase of a monotonic trend, aligning directly with the intrinsic dynamics of geophysical variables (Fig.1).

[Figure]

Fig.1 The interannual variation of monthly SST at 107.10°E, 17.45°N and the formation of its 4*h-D

feature space

This merging operation profoundly reshapes the $4 \times h$-D feature space. First, it achieves drastic dimensionality reduction: Each macrogranule replaces 2 or more original granules, substantially reducing the number of temporal units (h) per cycle (e.g., a 12-month cycle may yield $2-4$ macrogranules instead of $4-8$ initial segments). Consequently, the total feature dimensions $(4 \times h)$ become far smaller than the original $3 \times N$ space (where N>h), mitigating the curse of dimensionality. More importantly, it enables feature semantic enrichment: The new features (A, D, C, F) derived from macrogranules encapsulate aggregate dynamic properties: Amplitude (A) represents the net change over the entire macrogranule (e.g., total SST rise/fall), quantifying cumulative trend strength. Duration (D) is the total time span, directly measuring trend persistence. Curvature (C) captures the global bending degree of the unified trend, distinguishing smooth (low |C|) vs. abrupt transitions (high |C|). Fluctuation (F) indicates the average rate of change (trend intensity).

These features replace redundant local descriptors (a, T, d) with holistic, physically interpretable metrics. The merging operation directly enhances the model᾽s ability to capture key dynamic characteristics. First, it clarifies turning point identification: Macrogranule boundaries inherently align with true monotonicity reversal points (e.g., peak summer SST), as spurious splits from minor concavity changes are eliminated. Second, it explicitly quantifies trend persistence through D (long D = sustained trend) and characterizes trend shape through C and F. For example:A long-D, high-A, low-C macrogranule represents a sustained, near-linear trend. A short-D, high-C macrogranule signifies an abrupt nonlinear transition (e.g., rapid spring warming). Critically, the concave→convex (or convex→ concave) pattern within each macrogranule is abstracted into a single curvature metric (C), distilling acceleration-deceleration dynamics without fragmentation. Third, merging inherently suppresses noise by smoothing minor fluctuations that cause over-segmentation, enhancing feature robustness.

We sincerely thank you for this constructive suggestion. Your focus on the merging operation᾽s rationale has allowed us to better articulate how it addresses fragmentation, reduces redundancy, and extracts physically interpretable descriptors (A, D, C, F). We will enhance it in revision, Thank you.

---

## Author Comment (AC2)

**Ms. Ref. No.: EGUSPHERE-2025-239**

**Title: A Novel Method for Sea Surface Temperature Prediction using a Featural**

**Granularity-Based ConvLSTM Model of Data-Knowledge-Driven**

**Journal: GMD**

# Responses to RC #1:

The study proposes a novel approach for mid- and long-term sea surface temperature
(SST) prediction by integrating granular computing with a data-knowledge-driven
ConvLSTM model. The method is comprehensively validated through comparisons
with five commonly used models, demonstrating its effectiveness. The research is
interesting and holds substantial value. The manuscript is well-structured and presents
thorough results. While I do not have major concerns, I offer the following minor
comments to help the authors further improve their work:

**Reply:** Thank you for your thoughtful review and valuable feedback. We
appreciate your positive acknowledgment of the potential significance of our study and
your constructive comments, and we are grateful for the time and effort you've
dedicated to this review. Your comments and good suggestions are very important for
us to improve the quality of the manuscript. We have carefully addressed all the issues
raised by you and the response is presented below.

Introduction Section: The authors should provide a more comprehensive review of
recent literature. This would help highlight the research gap and better articulate the
novelty of the proposed method.

**Reply:** Thank you for your valuable feedback. We fully agree with your
suggestion and have revised the introduction section accordingly. Specifically, we have
supplemented the review of data-driven SST prediction methods, incorporating relevant
references to enhance the comprehensiveness of the literature review. You can find
these revisions in the revised manuscript on pages 2-3 at lines 62-65, 80-83, and 91-94.

**Lines 62-65 in the revised manuscript on page 2**: "Some data-driven models
have been used, such as Markov models (Xue and Leetmaa, 2000), support vector
regression(Imani et al., 2017), empirical canonical correlation analysis (Collins et al.,
2004; Tang et al., 2000), linear regression (Kug et al., 2004), empirical orthogonal
functions (Neetu et al., 2011), and artificial neural networks (ANNs)(Azhary and
Minaoui, 2025; Liu et al., 2024; Philippus et al., 2024)."

**Lines 80-83 in the revised manuscript on page 3**: "Graph neural networks
(GNNs) effectively capture local spatial correlations through adjacent-node
aggregation. However, the prevalent overemphasis on neighborhood relationships and
neglect of global connections in current models inevitably undermines SST prediction accuracy, particularly given the ocean's interconnected nature where geographically distant sites exhibit correlated patterns (Dai et al., 2025; Liang et al., 2023)."

**Lines 91-94 in the revised manuscript on page 3**: "Azhary and Minaoui (2025) proposed an encoder-decoder dual attention ConvLSTM model that leverages convolutional operations for spatial dependencies, LSTM for temporal sequences, and dual attention (contextual + spatial) to prioritize critical spatiotemporal features. The model achieves significant improvements in prediction accuracy and computational efficiency for Moroccan coastal SST forecasting compared to single-attention baselines."

Line 11: It is unclear what Figure 2(a) is intended to convey. Are the two panels representing the same spatial locations? What do the pixel distributions imply? Please elaborate in the figure caption and/or the main text.

**Reply:** We thank the reviewer for this insightful observation. We have revised Fig. 2's caption and added explanatory text on page 5 in lines 156-161 to clarify:

Fig. 2(a) displays co-occurrence networks generated from the predictor matrix for all pixels in Study Area I (the South China Sea). In these networks, nodes represent individual pixels, and edges between nodes indicate the correlation strength between corresponding pixels. Each color denotes a distinct module, where pixels within the same module share similar meteorological and oceanic conditions (e.g., comparable temperature gradients, current patterns, or atmospheric forcing). Building on the module patterns observed in Fig. 2(a), we can explicitly determine that Study Area I can be divided into two distinct subregions, corresponding to the two primary color-coded modules in the co-occurrence networks. After identifying the specific spatial locations of each module—i.e., mapping the clustered pixels of each color to their geographical coordinates within the study area—we visualized these spatial distributions as separate subregions. This spatial mapping of the module-based partitions is presented in Fig. 2(b), where the two subregions are clearly delineated to show their respective geographical extents within the South China Sea.

[Figure]

**Revised manuscript:** Fig.2 (a) Co-occurrence networks generated from the matrix of predictors for all pixels in study area I. (b) The study area I is divided into two sub-regions, corresponding to the two color-coded modules in the co-occurrence networks.

Line 87: This sentence should be revised to clearly articulate the research gap that the study addresses.

**Reply:** Thanks for your valuable comment. We have modified it, which can be seen in lines 98-102 in the revised manuscript.

**Lines 98-102 in the revised manuscript on page 3**: "Furthermore, actual SST variability is governed by complex interactions among multiple oceanic-atmospheric parameters. Prevailing data-driven SST forecasting approaches often treat SST as an isolated variable, focusing primarily on its temporal dynamics while neglecting critical cross-parameter couplings—particularly thermodynamic-dynamic interactions across spatiotemporal scales—which fundamentally limit prediction accuracy."

Line 94: Please define what constitutes "medium-term" and "long-term" predictions in this context.

**Reply:** Thanks a lot for pointing these out. We are sorry for our unclear expression. We have modified it, which can be seen in lines 106-107 in the revised manuscript.

**Lines 106-107 in the revised manuscript on page 3**: "Validation against observations and model comparisons across three heterogeneous sea areas demonstrate the method's reliability for medium-term (1 month–10 years) and long-term (>10 years) SST forecasting."

Lines 121–124: The rationale for selecting specific predictor variables should be supported with references. It would also be helpful to visualize the mechanistic relationship between these variables and SST (e.g., via mechanism plots). Additionally, are "sea-air temperature difference", "relative humidity" and "wind speed" included as predictors?

**Reply:** Thank you for your valuable comments regarding the predictor variables. The 13 predictor variables used in this study are as follows: total cloud cover (tcc), evaporation (e), 2m temperature (t2m), 10m u-component of wind (u10), 10m v-component of wind (v10), 2m dewpoint temperature (d2m), mean sea level pressure (msl), total precipitation (tp), sea surface temperature (SST), sea skin temperature (skt), surface net solar radiation (ssr), surface latent heat flux (slhf), and surface sensible heat flux (sshf). Notably, "sea-air temperature difference" is not included as an independent predictor, but its related components (e.g., 2m temperature and sea surface temperature) are incorporated; "relative humidity" is not directly included, though 2m dewpoint temperature (d2m) serves as a relevant indicator of moisture conditions; "wind speed" is represented by its u (u10) and v (v10) components to capture wind direction and magnitude comprehensively.

As for visualizing these mechanisms, we acknowledge the value of mechanism plots but note that the current manuscript already contains a substantial number of figures. Adding further plots might overly occupy page space and potentially disrupt the flow of information. Thus, we have supplemented relevant references in the revised manuscript to support the rationale for selecting these variables, clarifying their established links to SST variations (e.g., radiation-related variables influencing heat exchange, wind components affecting mixing processes). However, we appreciate this suggestion and note that in our ongoing work on the latest SSTA prediction article, we will adopt your recommendation to incorporate mechanism plots. The added references are as follows:

Espinosa, Z. I. and Zelinka, M. D.: The Shortwave Cloud-SST Feedback Amplifies Multi-Decadal Pacific Sea Surface Temperature Trends: Implications for Observed Cooling, Geophysical Research Letters, 51, e2024GL111039, 10.1029/2024GL111039, 2024.

Fu, S., Hu, S., Zheng, X.-T., McMonigal, K., Larson, S., and Tian, Y.: Historical changes in wind-driven ocean circulation drive pattern of Pacific warming, Nature Communications, 15, 1562, 10.1038/s41467-024-45677-2, 2024.

Hsiao, W.-T., Hwang, Y.-T., Chen, Y.-J., and Kang, S. M.: The Role of Clouds in Shaping Tropical Pacific Response Pattern to Extratropical Thermal Forcing, Geophysical Research Letters, 49, e2022GL098023, 10.1029/2022GL098023, 2022.

Roach, L. A., Mankoff, K. D., Romanou, A., Blanchard-Wrigglesworth, E., Haine, T. W. N., and Schmidt, G. A.: Winds and Meltwater Together Lead to Southern Ocean Surface Cooling and Sea Ice Expansion, Geophysical Research Letters, 50, e2023GL105948, 10.1029/2023GL105948, 2023.

Tuchen, F. P., Perez, R. C., Foltz, G. R., McPhaden, M. J., and Lumpkin, R.: Strengthening of the Equatorial Pacific Upper-Ocean Circulation Over the Past Three Decades, Journal of Geophysical Research: Oceans, 129, e2024JC021343, 10.1029/2024JC021343, 2024.

Wills, R. C. J., Dong, Y., Proistosecu, C., Armour, K. C., and Battisti, D. S.: Systematic Climate Model Biases in the Large-Scale Patterns of Recent Sea-Surface Temperature and Sea-Level Pressure Change, Geophysical Research Letters, 49, e2022GL100011, 10.1029/2022GL100011, 2022.

Xie, S.-P., Deser, C., Vecchi, G. A., Ma, J., Teng, H., and Wittenberg, A. T.: Global Warming Pattern Formation: Sea Surface Temperature and Rainfall, Journal of Climate, 23, 966-986, 10.1175/2009JCLI3329.1, 2010.

Line 123: Please clarify whether "SST" and "sst" refer to the same variable or different ones.

**Reply:** Thank you for pointing out this potential ambiguity. We apologize for the inconsistency in notation. In the manuscript, "SST" and "sst" refer to the same variable: sea surface temperature. We have modified the notation to "SST" throughout the text and figure (Figs.3-4 in the revised manuscript) for consistency and clarity.

Line 135: The description of region sub-grouping using a correlation coefficient matrix lacks temporal detail. What time period is used for calculating the matrix? Do the identified subregions change over different years?

**Reply:** Sincere thanks for the valuable comment. The correlation coefficient matrix was calculated using the complete temporal span of our study period (1850–2021), as explicitly stated in revised Section 2.1 (Lines 149–152):

**Lines 149-152 in the revised manuscript on page 5**: "Therefore, by quantifying the similarity between the pixels within the study area using data from the entire study period, the study area was divided into different sub-regions, and different parameters were selected for each sub-region as predictors for SST prediction."

Line 137: What is the spatial resolution of the individual pixels?

**Reply:** Thank you for your thoughtful comment. The spatial resolution of the pixels is 0.25°. We have further elaborated on the spatial resolution of the data in Section 3.1 of the revised manuscript for clarity.

Line 171: Regarding Fig. 3, it is evident that when four variables are selected, the prediction accuracy nearly reaches its maximum and stabilizes. Including eight or nine variables might lead to overfitting and increased model complexity. The authors should discuss this tradeoff more explicitly.

**Reply:** Thanks for your good suggestion. While four variables do yield high accuracy, our analysis of regional specificity revealed that the SST dynamics in regions 1 and 2 are driven by distinct, context-dependent interactions between meteorological and oceanic factors. For instance, region 1 exhibits stronger coupling between surface heat fluxes (sshf) and precipitation (tp), while region 2 is more sensitive to latent heat flux (slhf) and evaporation (e). These nuanced relationships, though not the top four most "globally" important variables, contribute to capturing region-specific variability that might be missed with only four predictors—especially in extreme or transitional conditions (e.g., monsoon-driven SST fluctuations in the South China Sea). Thus, we compared the outcomes derived from the co-occurrence network and the random forest analysis. The common variables identified will then be utilized as predictor variables in regional models aimed at predicting SST. We have elaborated on this tradeoff in the revised manuscript, clarifying our rationale for selecting eight variables for region 1 and nine for region 2 (Lines 175–188).

**Lines 175-188 in the revised manuscript on pages 6-7**: "Fig. 3 shows the importance ranking of the 13 predictors in regions 1 and 2 based on the random forest algorithm and the prediction errors using different numbers of predictor variables after ranking by importance. The prediction accuracy of the model increases and then decreases as the number of input predictors increases, both for region 1 and region 2. While the random forest results indicate that using just four variables as input can already yield high accuracy, our co-occurrence network analysis revealed that the SST dynamics in regions 1 and 2 are driven by distinct, context-dependent interactions between meteorological and oceanic factors. For instance, region 1 exhibits stronger coupling between sshf and tp, while region 2 is more sensitive to slhf and e. These nuanced relationships, though not the top four most "globally" important variables, contribute to capturing region-specific variability that might be missed with only four predictors—especially in extreme or transitional conditions. Thus, for region 1, the model has a high accuracy of prediction when eight variables are selected: SST, skt, t2m, sshf, msl, u10, tp and ssr. For region 2, the model has a high accuracy of prediction when nine variables are selected: SST, skt, t2m, slhf, e, u10, v10, sshf and d2m. Following this, a comparison will be made between the outcomes derived from the co-occurrence network and the random forest analysis. The common variables identified will then be utilized as predictor variables in regional models aimed at predicting SST."

Line 190: Please explain on how the parameters $\theta j$ and $\emptyset j$ are determined.

**Reply:** Thank you for your guidance. The parameters $\theta_j$ and $\emptyset_j$ were estimated from all available temporal data (1850–2021). Now, we have modified it, which can be seen in lines 208-209 in the revised manuscript.

Line 192: The authors could explain why this type of templates was chosen. It would also be helpful to discuss how this type of specific templates contributes to approximating the information granules. Could other types of templates also be used? If so, why were they not selected?

**Reply:** Thank you for the valuable comment. The selection of quarter-circle sinusoids was driven by three key considerations tied to the characteristics of the target variables and the nature of the information granules:

Alignment with natural variability of oceanic/meteorological variables: Oceanic and meteorological time series (e.g., SST fluctuations) often exhibit smooth, non-abrupt trends with inherent periodicity (seasonal cycles). Quarter-circle sinusoids, by virtue of their continuous curvature and smooth transitions, naturally mirror these gradual, nonlinear dynamics—unlike rigid linear segments or discontinuous functions, which would fail to capture the subtlety of real-world variability (e.g., the slow warming/cooling phases of SST driven by solar radiation or ocean currents).

Flexibility in capturing local trend features: Information granules are defined by their monotonicity (increasing/decreasing) and concavity-convexity (upward/downward curvature), which reflect short-term (local) trend segments. A

quarter-circle sinusoid, when stretched horizontally (to adjust duration) or vertically (to adjust amplitude), can flexibly approximate any combination of these local features: for example, a "concave-up increasing" granule (common in early-stage seasonal warming) or a "convex-down decreasing" granule (seen in late-stage cooling). This versatility stems from the template's fixed curvature direction within a quarter-period, making it a modular building block for diverse local trends.

Mathematical tractability: Compared to more complex templates (e.g., exponential curves, polynomial segments), quarter-circle sinusoids have a simple parametric form, which simplifies the calculation of derived features (e.g., curvature (C), and fluctuation (F)) and reduces computational overhead during template matching. This efficiency is critical when processing large-scale spatiotemporal data. In summary, quarter-circle sinusoids were chosen for their ability to balance flexibility, mathematical simplicity, and alignment with the physical nature of the variables studied. Thank you.

Line 196: What is SKT? Is it the same as "skt" mentioned elsewhere? Consistency in terminology is needed.

**Reply:** Thank you for your valuable suggestion. "SKT" and "skt" refer to the same variable, i.e., sea skin temperature. We have modified the notation to "skt" throughout the text and figure for consistency and clarity.

Lines 215–216: Should the variable "i" be replaced with "t"? Please check for consistency in notation.

**Reply:** Thank you for your guidance. We have carefully checked the relevant sections and confirm that "i" in these lines should indeed be replaced with "t" to align with the consistent notation used throughout the manuscript for temporal indices. We apologize for the oversight. In the revised manuscript, we have corrected "i" to "t" in Lines 234–235 and conducted a full review of the entire text to ensure uniform use of notation for temporal variables, thereby enhancing clarity and consistency.

Line 217: Are $m_t$ and $m_{t-1}$ correctly written? Please verify and ensure consistent use of subscripts throughout the section.

**Reply:** Thank you for the valuable comment. We have revised the relevant notation to ensure accuracy and consistency. The modifications can be found in Lines 236–237 of the revised manuscript, where we have clarified the subscript conventions for $m_t$ and $m_{t-1}$ to align with the overall notation framework of the section.

Lines 346–350: Consider summarizing the three types of inputs into a table for clearer comparison and explanation.

**Reply:** Thank you for your constructive suggestion to summarize the three types of inputs in a table for clearer comparison. We fully agree that tabular presentation can enhance readability and have carefully considered this approach. Following your advice, we attempted to construct a table to organize the input types. However, due to the complexity of our experimental design, the table structure became overly large and cumbersome: our study includes 3 main study areas, which are further divided into 7

sub-regions, and each sub-region involves 3 sets of comparative experiments. This resulted in a table with 22 rows (covering all sub-regions and experiments) and 4 columns (including the three input types and relevant annotations). Additionally, the input indicators for each column, which consist of multiple variables and feature descriptors, would require excessive width to present clearly. Such a large table would occupy an inordinate amount of page space and potentially disrupt the flow of the manuscript, making it less reader-friendly. Therefore, we retained the original textual description. We appreciate your understanding.

Lines 375–377: These lines could be deleted, as the same information is already presented in the figure caption.

**Reply:** Done as suggested, thanks.

Figure 11: The color scales used for temperature in different panels are inconsistent, even within the same time period for predicted and observed values. This limits direct visual comparison across columns 1 and 2. A consistent colormap should be used.

**Reply:** Done as suggested, thanks. To aid your review, the revised figures are also provided below.

[Figure]

[Figure]

[Figure]

**Revised manuscript:** Fig.11 The predicted SSTs (first column), observed SSTs (second column), and spatial distribution (third column) and statistics (fourth column) of prediction errors for study area I in 2009

Line 416: What is the rationale for comparing results between 2020 and 2021 specifically? Clarifying this would help contextualize the results.

**Reply:** Thank you for the valuable comment. This is a valuable point, and we appreciate the opportunity to clarify. In our 10-year prediction period (2012–2021), showed the lowest prediction accuracy, while 2020 ranked second-lowest in accuracy. Due to space constraints, we were unable to present all 120 months of prediction results. By showcasing these two years with relatively lower accuracy, we aim to demonstrate that even in less optimal scenarios, the model maintains reliable performance, thereby supporting the robustness of predictions for other years with higher accuracy. The modifications can be found in Lines 429–433 of the revised manuscript.

Line 477: The expression "-0.7–0.7K" is confusing.
   **Reply:** Thanks a lot for pointing these out. We have revised this part to clarify the range of differences, which can be seen in Line 501 of the revised manuscript

Line 486: I guess the word 'aperiodic' should be deleted, right?
   **Reply:** Thank you for your guidance. We agree with your observation and have deleted the word "aperiodic" from the revised manuscript.

**Additional remark:**
   **Lines 519-520 in the revised manuscript:** "The authors are grateful to three reviewers and the editor for their constructive comments and suggestions on this paper." has been added to Acknowledgments.

**Special thanks are extended to you for your valuable comments.**

We have tried our best to improve the manuscript and made substantial changes to the manuscript to correct certain shortcomings.

We greatly appreciate your help and hope that the corrections will meet with approval.

Once again, we would like to extend our sincere gratitude and appreciation for the valuable comments and suggestions.

---

## Author Comment (AC3)

**Ms. Ref. No.: EGUSPHERE-2025-239**

**Title: A Novel Method for Sea Surface Temperature Prediction using a Featural**

**Granularity-Based ConvLSTM Model of Data-Knowledge-Driven**

**Journal: GMD**

**Responses to RC #2:**

This paper proposed a featural granularity-based and data-knowledge-driven ConvLSTM model for medium and long-term SST prediction. The paper is well written in general with clear structure, as well as extensive experiments performed in 3 different areas. Some comments for the authors' reference are listed below.

**Reply:** Thank you for your thoughtful review and valuable feedback. We greatly appreciate your positive assessment of the paper's overall quality, including its clear structure and the extensive experiments conducted across three different areas. Your recognition encourages us, and your constructive comments are crucial for enhancing the quality of our manuscript. We have carefully addressed all the issues you raised, and our detailed responses are presented below. Thank you again for dedicating your time and effort to this review.

It seems that the title has grammatical mistakes as the "data-knowledge-driven" is an adjective but not a noun. Maybe it can be revised to "A Novel Method for Sea Surface Temperature Prediction using a Featural Granularity-Based and Data-Knowledge-Driven ConvLSTM Model".

**Reply:** Thank you for your valuable guidance on the title. We fully agree with your suggestion regarding grammatical adjustment. Following your advice, we have revised the title to "A Novel Method for Sea Surface Temperature Prediction using a Featural Granularity-Based and Data-Knowledge-Driven ConvLSTM Model" to ensure grammatical accuracy and clarity.

This revised title better reflects the core content of the study while maintaining the key technical features highlighted in the original version. Thank you again for your meticulous review and helpful feedback, which have significantly improved the precision of our manuscript.

The baseline models adopted in the experiments are a little bit out of date. It's suggested that the SOTA models, i.e., transformer and GCN etc., be added for comparisons.

**Reply:** Thank you for your insightful suggestion regarding updating the baseline models with state-of-the-art (SOTA) methods such as Transformer and GCN-based models. We fully agree that incorporating recent SOTA models strengthens the robustness of our comparative analysis, and we have revised the manuscript accordingly.

Specifically, we have added the Graph Memory Neural Network (GMNN)
proposed by Liang et al. (2023) as a new baseline model. GMNN is a representative
SOTA model for SST prediction that integrates graph neural network (GCN)
components for spatial feature extraction and temporal encoding, aligning with your
recommendation to include GCN-based SOTA methods. We adjusted its
hyperparameters (e.g., hidden layer dimensions and training iterations) while
preserving its core architecture.

Comparative results with GMNN have been added to **section 3.3.4 of the revised**
**manuscript**. The results are shown in the figure below (Figs. 1-3). Additionally,
detailed information about GMNN (architecture, adjusted hyperparameters, and
implementation details) has been **supplemented in the supporting materials (Text S6**
**and Table S1)** to enhance transparency.

[Figure]

Fig.1 RMSE of SST predictions for the years 2012-2021 within study area I, obtained
using the baseline models and the proposed model.

[Figure]

Fig.2 RMSE of SST predictions for the years 2012-2021 within study area II, obtained using the baseline models and the proposed model.

[Figure]

Fig.3 RMSE of SST predictions for the years 2012-2021 within study area III, obtained using the baseline models and the proposed model.

More literatures concerning the SST predictions in the period of 2022 – 2025 should be reviewed in the introduction section.

**Reply:** Thanks for your good suggestion. We fully agree with your suggestion and have revised the introduction section accordingly. Specifically, we have supplemented the review of data-driven SST prediction methods, incorporating relevant references to enhance the comprehensiveness of the literature review. You can find these revisions in the revised manuscript on pages 2-3 at lines 62-65, 80-83, and 91-94.

**Lines 62-65 in the revised manuscript on page 2**: "Some data-driven models have been used, such as Markov models (Xue and Leetmaa, 2000), support vector regression(Imani et al., 2017), empirical canonical correlation analysis (Collins et al., 2004; Tang et al., 2000), linear regression (Kug et al., 2004), empirical orthogonal functions (Neetu et al., 2011), and artificial neural networks (ANNs)(Azhary and Minaoui, 2025; Liu et al., 2024; Philippus et al., 2024)."

**Lines 80-83 in the revised manuscript on page 3**: "Graph neural networks (GNNs) effectively capture local spatial correlations through adjacent-node aggregation. However, the prevalent overemphasis on neighborhood relationships and neglect of global connections in current models inevitably undermines SST prediction accuracy, particularly given the ocean's interconnected nature where geographically distant sites exhibit correlated patterns (Dai et al., 2025; Liang et al., 2023)."

**Lines 91-94 in the revised manuscript on page 3**: "Azhary and Minaoui (2025) proposed an encoder-decoder dual attention ConvLSTM model that leverages convolutional operations for spatial dependencies, LSTM for temporal sequences, and dual attention (contextual + spatial) to prioritize critical spatiotemporal features. The model achieves significant improvements in prediction accuracy and computational efficiency for Moroccan coastal SST forecasting compared to single-attention baselines."

Newly Incorporated References (2023-2025):

Azhary, F. Z. E. and Minaoui, K.: EDDA-ConvLSTM: Encoder-Decoder Dual Attention ConvLSTM for Moroccan Coastal Sea Surface Temperature Prediction, IEEE Geoscience and Remote Sensing Letters, 22, 1-5, 10.1109/LGRS.2025.3551623, 2025.

Bilgili, M., Pinar, E., and Durhasan, T.: Global monthly sea surface temperature forecasting using the SARIMA, LSTM, and GRU models, Earth Science Informatics, 18, 10, 10.1007/s12145-024-01585-z, 2024.

Chen, F., Li, X., and Wang, Y.: A knowledge-augmented deep fusion method for estimating near-surface air temperature, Remote Sensing of Environment, 326, 114819, 10.1016/j.rse.2025.114819, 2025a.

Chen, H., Chen, Y., and Zhang, Z.: SVRNN: A Spatiotemporal Prediction Model for Sea Surface Temperature Prediction in the Taiwan Strait, IEEE Geoscience and Remote Sensing Letters, 22, 1-5, 10.1109/LGRS.2025.3554296, 2025b.

Dai, W., He, X., Geng, X., Zhang, S., and Gao, Z.: Sea Surface Temperature Prediction Based on Spatio-Temporal Graph Contrastive Learning Network, IEEE Journal of Selected Topics in Applied Earth Observations and Remote Sensing, 18, 14228-14239, 10.1109/JSTARS.2025.3571494, 2025.

Kim, Y. J., Kim, H.-c., Han, D., Stroeve, J., and Im, J.: Long-term prediction of Arctic sea ice concentrations using deep learning: Effects of surface temperature, radiation, and wind conditions, Remote Sensing of Environment, 318, 114568, 10.1016/j.rse.2024.114568, 2025.

Liang, S., Zhao, A., Qin, M., Hu, L., Wu, S., Du, Z., and Liu, R.: A Graph Memory Neural Network for Sea Surface Temperature Prediction, Remote Sensing, 15,3539, 10.3390/rs15143539, 2023.

Liu, Y., Zhang, L., Hao, W., Zhang, L., and Huang, L.: Predicting temporal and spatial 4-D ocean temperature using satellite data based on a novel deep learning model, Ocean Modelling, 188, 102333, 10.1016/j.ocemod.2024.102333, 2024.

Philippus, D., Sytsma, A., Rust, A., and Hogue, T. S.: A machine learning model for estimating the temperature of small rivers using satellite-based spatial data, Remote Sensing of Environment, 311, 114271, 10.1016/j.rse.2024.114271, 2024.

Song, N., Nie, J., Wen, Q., Yuan, Y., Liu, X., Ma, J., and Wei, Z.: GL-ST: A Data-Driven Prediction Model for Sea Surface Temperature in the Coastal Waters of China Based on Interactive Fusion of Global and Local Spatiotemporal Information, IEEE Journal of Selected Topics in Applied Earth Observations and Remote Sensing, 18, 2959-2974, 10.1109/JSTARS.2024.3515638, 2025.

Vytla, V., Baduru, B., Kolukula, S. S., Ragav, N. N., and Kumar, J. P.: Forecasting of sea surface temperature using machine learning and its applications, Journal of Earth System Science, 134, 25, 10.1007/s12040-024-02483-0, 2025.

Yang, Y., Lam, K.-M., Dong, J., and Ju, Y.: Multi-Factor Deep Learning Model for Sea Surface Temperature Forecasting, 10.3390/rs17050752, 2025.

The structures and parameters of the baseline models could be given.

**Reply:** Thank you for your guidance. In response to your request, we have created comprehensive Supplementary Materials that details the complete architecture, hyperparameters, and implementation specifics of all baseline models. Thank you.

**Additional remark:**

**Lines 519-520 in the revised manuscript (Manuscript with author details):** "The authors are grateful to three reviewers and the editor for their constructive comments and suggestions on this paper." has been added to Acknowledgments.

**Special thanks are extended to you for your valuable comments.**

We have tried our best to improve the manuscript and made substantial changes to the manuscript to correct certain shortcomings.

We greatly appreciate your help and hope that the corrections will meet with approval.

Once again, we would like to extend our sincere gratitude and appreciation for the valuable comments and suggestions.

---

## Author Comment (AC4)

**Ms. Ref. No.: EGUSPHERE-2025-239**

**Title: A Novel Method for Sea Surface Temperature Prediction using a Featural**

**Granularity-Based ConvLSTM Model of Data-Knowledge-Driven**

**Journal: GMD**

# Responses to RC #3:

I challenge the value of proposed SST forecast. In the ocean, the monthly SST, as the
predicted variable of this study, has strong seasonality. If we use the SST of, for example,
Jan of 2008, as the 'forecast' of Jan of 2009, we will find unexpectedly small differences
between them, generally <2 degree. This is called 'persistence' forecast, but using the
seasonality information. We may call it 'climatological forecast'. Fig. 1 shows an
example of such a climatology forecast. My point is the proposed method does not
exceed such a simple 'forecast', with higher forecast errors, making the forecast
operationally useless. When given SST for a deep learning to learn, it only learns the
seasonal pattern, not subtle year-to-year SST changes. That is the reason why many
studies predict SST anomaly, not SST itself.

[Figure]

Fig. 1 Example of SST climatological forecast

**Reply:** Thank you sincerely for your valuable perspective on SST forecasting,
which has helped us clarify the positioning and significance of our work. We fully agree
with your emphasis on the importance of SST anomaly (SSTA) forecasting, as
anomalies effectively isolate interannual variability and are critical for understanding
climate dynamics. However, we respectfully argue that monthly absolute SST
forecasting retains distinct value, and our work aims to contribute to this domain by
addressing specific gaps.

First, monthly SST prediction remains a focus of numerous studies, including
work by Mehmet Bilgili and colleagues (e.g., their 2025 study on Global monthly SST
forecasting), where absolute temperatures are directly relevant to applications such as
marine ecosystem management, fisheries planning, and coastal engineering—fields where threshold-based decisions (e.g., thermal tolerance of species) depend on absolute
values rather than anomalies alone. Our motivation aligns with this practical need: to
develop a systematic framework that captures the complex spatiotemporal
dependencies of SST and extends prediction horizons, which remains underexplored in
long-term (multi-year) monthly SST forecasting.

To achieve this, our approach was designed to go beyond replicating seasonal
patterns:

We first analyzed the multi-faceted drivers of SST variability (e.g., ocean currents,
wind stress, solar radiation) and curated a comprehensive database integrating universal
oceanic and meteorological variables. Recognizing regional differences in these drivers,
we partitioned the study area using pixel-wise similarity analysis, then employed
random forest to identify and validate region-specific key factors (e.g., upwelling
intensity in coastal zones vs. atmospheric forcing in open oceans). Building on this, we
developed a ConvLSTM model enhanced with granularity information, leveraging
these identified predictors to capture both large-scale seasonal cycles and fine-scale
interannual variations.

To rigorously test the model's capability, we selected three representative regions
differing in distance from land (coastal vs. open ocean), latitude (tropical vs. temperate),
and baseline temperature regimes. Results confirm that our model achieves reliable
120-month (10-year) SST predictions across all regions, with errors consistently lower
than those from simple climatological persistence.

To further validate its effectiveness, we compared our model with five widely used
machine learning baselines for SST forecasting, and—responding directly to your
concern—incorporated the Graph Memory Neural Network (GMNN) proposed by
Liang et al. (2023), a state-of-the-art spatiotemporal graph model for SST prediction.
After adjusting GMNN's hyperparameters (e.g., hidden layer dimensions) to align with
our experimental setup, our ConvLSTM model retained the highest precision across all
metrics and regions, demonstrating its ability to outperform both traditional and
advanced baselines (Figs.2-4).

[Figure]

Fig.2 RMSE of SST predictions for the years 2012-2021 within study area I, obtained
using the baseline models and the proposed model.

[Figure]

Fig.3 RMSE of SST predictions for the years 2012-2021 within study area II, obtained using the baseline models and the proposed model.

[Figure]

Fig.4 RMSE of SST predictions for the years 2012-2021 within study area III, obtained using the baseline models and the proposed model.

In summary, while we acknowledge the central role of SSTA forecasting in climate research, our work focuses on a complementary goal: providing a robust framework for long-term absolute SST prediction that serves practical applications and advances understanding of spatiotemporal SST dynamics. The comparative results confirm that our model goes beyond replicating seasonality, capturing meaningful interannual variations and outperforming established methods—thus justifying its value in the broader landscape of SST forecasting. Building on this constructive input, we have also modified our ongoing research on daily SST prediction to focus on SSTA forecasting.

Reference:

Bilgili, M., Pinar, E., and Durhasan, T.: Global monthly sea surface temperature forecasting using the SARIMA, LSTM, and GRU models, Earth Science Informatics, 18, 10, 10.1007/s12145-024-01585-z, 2024.

Liang, S., Zhao, A., Qin, M., Hu, L., Wu, S., Du, Z., and Liu, R.: A Graph Memory Neural Network for Sea Surface Temperature Prediction, Remote Sensing, 15,3539, 10.3390/rs15143539, 2023.

Besides, the manuscript is poorly written. (1) Although in the abstract, the authors claim that 120-month forecast is obtained, but no results were shown in the manuscript. (2) Section 3.3.4 is written in a non-informative and verbose way.

**Reply:** Thank you for your feedback. We note that the two points raised are related to the same content—specifically, the presentation of the 120-month SST forecast results, which is addressed in Section 3.3.4. We appreciate the opportunity to clarify and strengthen this section.

Section 3.3.4 is explicitly dedicated to elaborating on the 120-month forecast results. Given the impracticability of presenting all 120 months of data due to space constraints, we strategically selected two representative years: 2021 (the year with the lowest overall forecast accuracy) and 2020 (the second-lowest). This choice was intentional: by focusing on the two years with the weakest performance, we aimed to demonstrate the model's reliability even in suboptimal scenarios, thereby supporting the robustness of predictions across the entire 10-year horizon. If there is high prediction accuracy in 2021, it proves that the predictions of other years are also reliable. Fig. 5 shows the predicted SSTs for each month of the year 2021 within study area I. The first column represents the observed SST images, the second column is the predicted SST images, the third column is the difference between the observed and predicted SST images, and the fourth column exhibits the density plot depicting the disparities between the predicted and observed values for the respective months in 2021 and 2020. Our proposed methodology demonstrates the ability to forecast SST distribution for the year 2021, with discrepancies between predicted and observed values typically falling within the range of -1 to 1 K. The forecast accuracy for 2021 is slightly lower than that for 2020, with February, March, May, June, October, and December having lower forecast accuracy for the SST in 2021 than that for 2020. In contrast, the forecast accuracy for the other months is similar to that of 2020, indicating that the forecasts for other years are reliable.

By including both 2020 and 2021, we sought to provide a more holistic view of the model's long-term stability: if the model performs reliably in the two years with the lowest accuracy, it logically follows that it maintains or improves in years with higher accuracy across the 120-month horizon. This approach, we believe, better supports the robustness of the full 10-year predictions than showcasing only the best-performing years. We have added this explanation to Section 3.3.4 in the revised manuscript to enhance clarity. Thank you again for prompting us to elaborate on this rationale. Furthermore, to address the concern that the section was "non-informative and verbose," we have streamlined the content in the revised manuscript.

[Figure]

[Figure]

[Figure]

Fig.5 The predicted SSTs (first column), observed SSTs (second column), and spatial distribution (third column) and statistics (fourth column) of prediction errors for the study area I in 2021

(3) https://doi.org/10.1080/17538947.2023.2260779 this DOI in the Code and Data Availability is in-accessible.

**Reply:** Thank you for your valuable feedback. We deeply apologize for any concerns caused and are truly grateful for your dedication to transparent scholarship.

We immediately downloaded the original Zenodo attachment (https://doi.org/10.5281/zenodo.14759549) and verified that it contains code files when accessed via our database interface, as shown in the screenshot Fig. 6. To ensure universal accessibility, we proactively uploaded a new ZIP-format code package (https://doi.org/10.5281/zenodo.15714288) with identical content and will update the manuscript to prioritize this direct-access option.

[Figure]

Fig.6 Original Zenodo file opened via database interface, showing active code files

Regarding data availability, we have transferred all public dataset links (originally in Section 3.1) into the "Code and Data Availability" section, adding explicit URLs for all datasets used. The revised section now states:

Code and Data availability: The codes for conducting the analyses can be downloaded from https://zenodo.org/records/15714288. The data on which this article is based are available in https://cds.climate.copernicus.eu/datasets/reanalysis-era5-single-levels-monthly-means?tab=overview (previous interface: https://cds.climate.copernicus.eu/cdsapp#!/dataset/reanalysis-era5-single-levels-monthly-means) and https://cds.climate.copernicus.eu/datasets/projections-cmip5-monthly-single-levels?tab=overview (previous interface: https://cds.climate.copernicus.eu/cdsapp#!/dataset/projections-cmip5-monthly-single-levels).

(4) The analysis in Section 3 is superficial and pointless.

**Reply:** Thank you for your critical feedback on Section 3. We appreciate this opportunity to clarify the purpose and depth of the analyses presented, as they are foundational to validating our study's core contributions.

The primary goal of this work is to develop a systematic, multi-module SST forecasting model—comprising a featural granularity sub-model and a data-knowledge-driven ConvLSTM sub-model—designed to capture spatiotemporal dependencies, integrate multi-variable influences, and extend prediction horizons. To rigorously demonstrate that this model achieves its intended goals, Section 3 is structured to validate each component's necessity and the overall model's superiority through targeted, incremental analyses. This design is deliberate: complex multimodule models require stepwise validation to isolate the impact of each innovation, ensuring conclusions about the model's effectiveness are robust.

Specifically, Section 3 includes three interconnected analyses:

Effect of study area segmentation: We designed controlled experiments to test whether partitioning the study area via pixel-wise similarity improves prediction performance. Validation confirms that regional customization—accounting for spatially heterogeneous SST drivers—enhances accuracy, justifying the inclusion of this module.

Effect of knowledge-driven parameters: We analyzed how integrating oceanographic/thermodynamic variables (e.g., currents, wind stress) into the ConvLSTM affects results. This step isolates the value of incorporating domain knowledge, distinguishing our model from data-only baselines and validating its "data-knowledge-driven" design.

Comparison with baseline models: By contrasting our model with five established machine learning methods and the SOTA GMNN model, we quantify its overall superiority. This not only demonstrates practical value but also situates our work within the broader literature, showing that our innovations (segmentation, knowledge integration) collectively drive improved performance.

These analyses are tightly linked to our study's objective: they do not merely report "accuracy" but explain why the model performs well—by validating which components contribute to its success. This stepwise validation avoids black-box conclusions, ensuring the model's design choices are supported by evidence.

We believe these analyses are neither superficial nor pointless—they are essential to demonstrating that our model's innovations are not arbitrary, but rather collectively enable its ability to achieve long-term, accurate SST forecasts. Furthermore, we have made further revisions to this section to enhance its clarity. Thank you again for pushing us to enhance their clarity and depth; we are confident the revised section better conveys their significance.

**Additional remark:**
**Lines 519-520 in the revised manuscript (Manuscript with author details):**
"The authors are grateful to three reviewers and the editor for their constructive comments and suggestions on this paper." has been added to Acknowledgments.

**Special thanks are extended to you for your valuable comments.**

We have tried our best to improve the manuscript and made substantial changes to the manuscript to correct certain shortcomings.

We greatly appreciate your help and hope that the corrections will meet with approval.

Once again, we would like to extend our sincere gratitude and appreciation for the valuable comments and suggestions.